# Effect of Catalyst Ink and Formation Process on the Multiscale Structure of Catalyst Layers in PEM Fuel Cells

Huiyuan Liu [1], Linda Ney [2], Nada Zamel [2,*] and Xianguo Li [1,*]

1    Department of Mechanical and Mechatronics Engineering, University of Waterloo,
     Waterloo, ON N2L 3G1, Canada; h577liu@uwaterloo.ca
2    Fraunhofer Institute for Solar Energy Systems ISE, 79110 Freiburg, Germany; linda.ney@ise.fraunhofer.de
*    Correspondence: nada.zamel@ise.fraunhofer.de (N.Z.); xianguo.li@uwaterloo.ca (X.L.)

**Abstract:** The structure of a catalyst layer (CL) significantly impacts the performance, durability, and cost of proton exchange membrane (PEM) fuel cells and is influenced by the catalyst ink and the CL formation process. However, the relationship between the composition, formulation, and preparation of catalyst ink and the CL formation process and the CL structure is still not completely understood. This review, therefore, focuses on the effect of the composition, formulation, and preparation of catalyst ink and the CL formation process on the CL structure. The CL structure depends on the microstructure and macroscopic properties of catalyst ink, which are decided by catalyst, ionomer, or solvent(s) and their ratios, addition order, and dispersion. To form a well-defined CL, the catalyst ink, substrate, coating process, and drying process need to be well understood and optimized and match each other. To understand this relationship, promote the continuous and scalable production of membrane electrode assemblies, and guarantee the consistency of the CLs produced, further efforts need to be devoted to investigating the microstructure of catalyst ink (especially the catalyst ink with high solid content), the reversibility of the aged ink, and the drying process. Furthermore, except for the certain variables studied, the other manufacturing processes and conditions also require attention to avoid inconsistent conclusions.

**Keywords:** PEM fuel cell; catalyst layer; multiscale structure; catalyst ink; ink formulation; ink deposition process

## 1. Introduction

With the increased energy crisis, environmental and climatic deterioration, as well as health risks, the use of clean renewable energy becomes important. However, due to the limitation of period and region, it is hard to guarantee the continuous supply for most renewable energy, e.g., solar or wind energy. Therefore, it is necessary to explore clean and efficient carriers to store or transport renewable energy, where hydrogen stands out due to zero-carbon emission. To effectively apply hydrogen in the energy economy, proton exchange membrane (PEM) fuel cells are expected to play a significant role. PEM fuel cells directly convert the chemical energy stored in hydron and oxygen to electrical energy through the oxidation of hydrogen at the anode and the reduction of oxygen at the cathode. If green hydrogen, commonly produced from renewable energy through water electrolysis, could be used as fuel, the by-products include only water and heat, significantly reducing zero-carbon emissions [1,2]. Furthermore, compared with other energy conversion technologies, e.g., internal combustion engines or other kinds of fuel cells, PEM fuel cells also have the advantages of high energy efficiency and low operation temperature [3,4]. After more than 30 years of research and development, PEM fuel cells have been successfully applied to stationary and potable generation systems, as well as to sea, land, air, or rail transportation, and are on their way to commercialization in some fields, e.g., vehicles. However, due to the low operation temperature, the use of expensive

platinum (Pt)-based catalysts is necessary to promote the electrode reactions to supply the required performance and durability for practical application [5]. Therefore, to compete with internal combustion engines and Li-ion batteries, the performance and costs must be reduced, while durability should be increased.

The catalyst layer (CL), where the electrode reactions occur, is a key component of PEM fuel cells. The CL typically constitutes the continuous and intersecting networks of carbon-supported Pt-based catalysts (supplying active sites and electron transport pathway), ionomers (supplying proton transport pathway and acting as a binder to enhance the mechanical stability of CLs), and pores (supplying reactants and water transport pathway). The electrochemical reactions occur at the intersection region of the three components, mostly referred to as the triple-phase boundary (TPB). Therefore, not only the activity of catalysts, but the accessibility of active sites to gaseous reactants, protons, and electrons, i.e., the formation of TPB, is also essential for improving the performance and durability of PEM fuel cells and thus decreasing the cost. This may be why, although the activity and durability of catalysts have been improved dozens of times in the last decade, the performance and durability of PEM fuel cells have only slightly increased [6–8]. In brief, the performance, durability, and cost of CLs, and hence PEM fuel cells, are largely dependent on the micro-, meso-, and macro-structures of CLs, which determine the electrochemical surface area (or TPB) as well as the effective transport properties of proton, electron, reactants, water, and heat.

One way to optimize the micro-, meso-, and macro-structures of CLs is the optimization of the manufacturing process of the conventional CLs; the other way is fabricating the CLs with ordered structure. Conventional CLs are primarily manufactured by a catalyst ink-based processing method. To create the TPB and the networks for mass transport, supported catalysts and ionomers are typically dispersed in a liquid chemical solvent, referred to as catalyst ink. The catalyst ink is then coated on a substrate surface to form an ink film, and the solvent in the ink film is evaporated through a drying process to form the solid structure of the CLs.

To mitigate the mass transport issues caused by the winding transport pathway in conventional CLs, the structure-ordered CL was first sketched by Middelman et al. [9], which is constituted of the aligned electronic conductor loaded with Pt nanoparticles perpendicular to the membrane, being covered by ~10 nm thick ionomer layer. This structure is expected to maximize the TPB and boost mass transport by allowing the transport pathway of proton, electron, reactants, and water to be as short as possible. Currently, structure-ordered CLs primarily have four types, based on the ordered electronic conductors (e.g., vertically aligned carbon nanotubes [10,11] or conductive polymer [12,13]), ordered catalysts [14–16], ordered ionomer materials [17], or ordered non-conductive materials (e.g., nanostructured thin film (NSTF) CLs [18]). Pt loading has been successfully decreased to less than 0.125 mg/cm$^2$ MEA (membrane electrode assembly, MEA) without sacrificing fuel cell performance in some structure-ordered CLs [16,18,19]. However, several disadvantages hampering their practical application in PEM fuel cells systems remain, such as complicated and costly fabrication processes, as well as severe water flooding in some cases owing to the hydrophilicity of ultrathin structures. The structure-ordered CLs are usually ultrathin, less than 1 μm or even less than 200 nm in some cases [14,16,20], which will result in inherently low mass transport resistance versus the conventional CLs with a thickness of several μm due to the shorter transport distance. However, most structure-ordered CLs are susceptible to topple over [14,18], thus losing their ordered structure during the decal or hot-pressing process. Since there is a long way needed for the practical application of structure-ordered CLs, this review focused on conventional CLs (hereafter simply referred to as CLs) practically applied at present.

The multiscale structure of CLs is formed during the manufacturing process and is thus influenced by the composition (or formulation), order of ingredient mixing, and preparation/mixing process of catalyst ink, the substrate, and the application (deposition or coating) process and drying process of catalyst ink (Figure 1). Recent studies indicate

that the composition and preparation process of the catalyst ink impact the multiscale structure of CLs to a large part, through affecting its microstructure and macroscopic properties [21–24]. The catalyst inks are biphasic material systems (suspensions) commonly composed of solid catalyst and ionomer as well as solvent. The type of catalyst, ionomer, and solvent, ionomer/carbon (I/C) ratio, or the solid content determine the underlying interactions between the individual constituents. These interactions will influence the microstructure of catalyst ink, e.g., the size and shape of primary catalyst–ionomer agglomerates, conformation, and size of primary ionomer agglomerates, agglomeration, or the interface of catalyst and ionomer [25–27]. Furthermore, to achieve a well-dispersed ink, an effective dispersion process is necessary to break up the initial large agglomerates into the desired size [28]. The ink formulation and preparation processes influence its macroscopic properties directly or indirectly through impacting its microstructure, e.g., rheology (viscosity or thixotropy), surface/interface tension, or stability, which would influence the coating quality or drying processes and thus the structure of CLs made of the ink [29–35]. Hence, understanding and tailoring the microstructure and macroscopic properties of the ink by controlling its formulation and preparation process, i.e., catalyst ink engineering, are significantly important in the formation of the optimum CL structure.

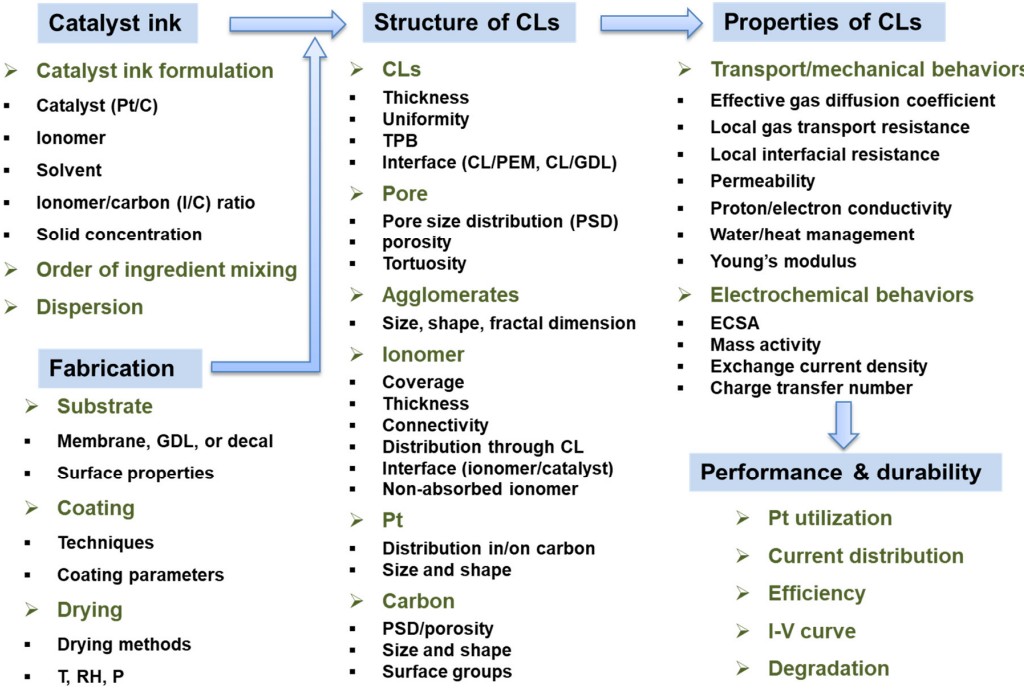

**Figure 1.** Material–structure–properties relation of the CL (T: temperature, RH: relative humidity, P: pressure, ECSA: electrochemical active surface area).

When the catalyst ink is prepared, the substrate and how the catalyst ink will evolve into the final CL during the coating and drying process also have a critical influence on the CL structure. The commonly used substrates include gas diffusion layer (GDL, carbon paper/cloth coated with microporous layer (MPL)), membrane, or decal film. The catalyst ink, substrate, coating process, and solvent evaporation should be elaborately adjusted to form an adherent and continuous CL [35,36]. For example, different coating techniques have different demands on the viscosity or the choice of solvent [37,38]. For spray coating, the ink with low solid content, hence low viscosity, is necessary to avoid the blockage of the spraying nozzle [36]. However, for screen printing, slot-die coating, or gravure coating, the viscosity of the catalyst ink needs to be increased [38,39]. After solvent evaporation, the deposited ink film evolves into a CL. During the solvent evaporation process, the movement, aggregation, or self-assembly of the agglomerates in the ink film occur, determining the final structure of the CL [40,41]. The solvent evaporation rate is

primarily impacted by the solvent properties or the drying methods and conditions, e.g., the boiling point and vapor pressure of solvent, temperature, or relative humidity (RH) [40–42].

Over the last 30 years, considerable effort has been devoted to catalyst ink engineering, development of novel coating methods, as well as optimization of the coating parameters and drying process, to maximize the TPB and minimize the mass transport resistance in CLs. However, the relationship of the recipe and preparation process of catalyst ink or the CL formation process and the multiscale structure of CL and further the performance and durability of PEM fuel cells is still unclear and even inconsistent in some cases.

For the time being, there have been some reviews focused on the optimization of the formulation and dispersion techniques of catalyst ink, the characterization and modeling of catalyst ink, the coating techniques, or the underlying interactions between solvent, ionomer, and catalyst [29,35,43]. However, there are few reviews focusing on the relationship between the composition and preparation process of the catalyst ink or the CL formation process and the multiscale structure of CL and further the performance and durability of PEM fuel cells. The relation between the multiscale structure of CL and the performance and durability of PEM fuel cells has been summarized in our previous review [44]; hence, this review focuses only on the effect of the formulation and preparation of catalyst ink on its microstructure and macroscopic properties and further on the CL structure as well as the effect of the CL formation process on the multiscale structure of CL. At first, brief overviews on the multiscale structure of CL and the microstructure and macroscopic properties of catalyst ink are presented. Then, it proceeds to the effect of catalyst type, ionomer, and solvent, or their ratios, as well as the order of ingredient mixing and dispersion process on the microstructure and macroscopic properties of catalyst ink and further on the multiscale structure of CL. After this, the CL formation process and its effect on the formation, morphology, or structure of CL are reviewed, including the selection of substrate and coating and drying processes. Finally, several perspectives on future research and development in this area are provided.

## 2. Multiscale Structure of CL in PEM Fuel Cell

The CL is a three-dimensional intersecting network of catalyst, ionomer, and pore. The electrochemical reaction in the CL occurs at the TPB, as described earlier. Hence, this requires the mass transport pathways to/from the TPB in CLs and hence demands the ionomer or water for proton transport, Pt/C for catalyzing the electrochemical reaction and electron transport, and the pores next to the fully or partially covered Pt/C for the transport of reactant gas and water. This is schematically illustrated in Figure 2.

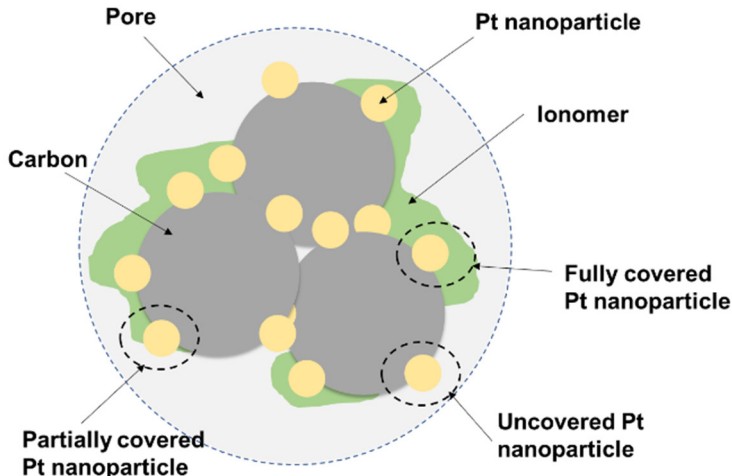

**Figure 2.** Illustration of ionomer-coated catalyst agglomerate [21,45].

Although typical CLs only have a thickness of 5–10 μm [21], they have a complicated multiscale structure, i.e., micro-, meso-, and macro-structure, due to the catalyst particles

and ionomer, primary agglomerate particles, and secondary aggregation of primary agglomerates. The catalyst particles, 2–5 nm Pt nanoparticles supported on 10–50 nm carbon black with a pore diameter of a few nm, tend to strongly bond together and aggregate, forming primary catalyst agglomerates, which are commonly surrounded by absorbed ionomer films 3.5–20 nm thick [46], i.e., primary catalyst–ionomer agglomerates of a few hundred nm [47]. The primary catalyst–ionomer agglomerates usually aggregate into large secondary agglomerates of a few μm through Van Der Waals force. Therefore, the secondary catalyst–ionomer agglomerates usually exhibit a bimodal pore size distribution, including 2–20 nm-sized pores within primary agglomerates and larger over 20 nm-sized pores between primary agglomerates in the secondary agglomerates—the former is often referred to as the primary pores, and the latter the secondary pores [47,48]. Moreover, there also exist large irregular ionomer patches formed by non-absorbed ionomer agglomerates [49]. The distribution of ionomer in CLs is usually inhomogeneous, in the sense that the coverage of Pt nanoparticles can be categorized as (i) fully covered with a thick ionomer film, (ii) fully covered with a thin ionomer film, (iii) partially covered by ionomer, and (iv) completely uncovered by ionomer [45]. Hence, to meet the activity requirements for high current production, the ionomer film distribution needs to be optimized.

Many advanced characterization techniques and simulation methods have been used to investigate the multiscale structure of CLs [44,50]. The studies of the CLs structure generally include but are not limited to (i) CL: the uniformity of the individual constituents in CL, the uniformity of CL thickness, surface roughness, TPB, or the interfaces of CL and membrane or CL and GDL; (ii) pore: pore size distribution, porosity, and tortuosity; (iii) agglomerates: the size, shape, or fractal dimension; (iv) ionomer: the ionomer coverage on Pt and carbon, thickness of ionomer film, ionomer connectivity, distribution of ionomer through the CL, the interaction/interface between ionomer and Pt, non-absorbed ionomer agglomerates in the macropores; (v) Pt: distribution in/on carbon, or size and shape; (vi) carbon: pore size distribution, porosity, size and shape, or surface groups (Figure 1). These would impact the physicochemical and electrochemical properties of CL and thus influence the activation, ohmic, or concentration polarization and durability of PEM fuel cells (Figure 1). The physicochemical properties mainly include effective diffusion coefficient, local gas transport resistance, local interfacial resistance between ionomer and Pt, or permeability (gas transport), capillary pressure and contact angle (water management), effective electron/proton/thermal conductivity, as well as Young's modulus, which govern the transport or mechanical behaviors of CL [44]. The electrochemical properties primarily include electrochemical surface area, mass activity, exchange current density, charge transfer number [44].

## 3. Microstructure and Macroscopic Properties of Catalyst Ink

### 3.1. Microstructure of Catalyst Ink

The CL is generally fabricated by a catalyst ink-based process, as described in the Section 1. A state-of-the-art catalyst ink is often prepared by uniformly mixing catalyst powder, ionomer, and solvent. Although being effectively dispersed, there still exist some undispersed primary and secondary agglomerates due to the complex interplay of interactions between catalyst, ionomer, and solvent. Taking the water/alcohol (1-propanol, isopropanol, or ethanol)-based catalyst ink commonly used as an example, primary catalyst agglomerates surrounded by ionomer (i.e., primary catalyst-ionomer agglomerates), non-absorbed primary ionomer agglomerates, and secondary agglomerates formed by primary agglomerates [21,51–55] (Figure 3) may exist. During the coating and drying process, these primary and secondary agglomerates assemble into a porous structure, forming the multiscale structure of CL [56]. Therefore, the multiscale structure of CL and hence its performance and durability would be governed by the size, size distribution, and shape of the various agglomerates, the absorbed ionomer content on the catalyst, or the interface of catalyst|ionomer in the catalyst ink, which are sensitive to the ink formulation and dispersion process [49,57–60] (Figure 4).

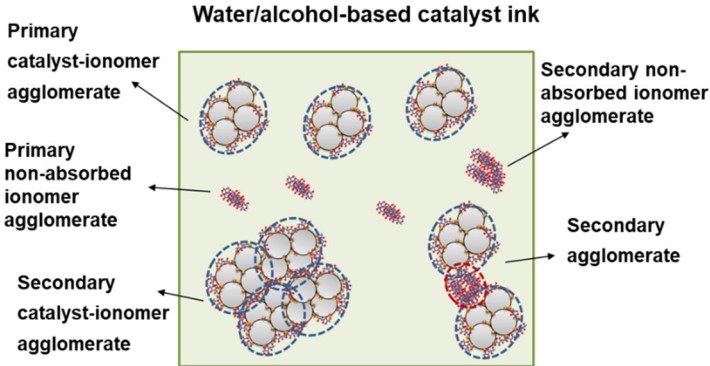

**Figure 3.** Illustration of the microstructure of the water/alcohol-based catalyst ink commonly used [21,51–55].

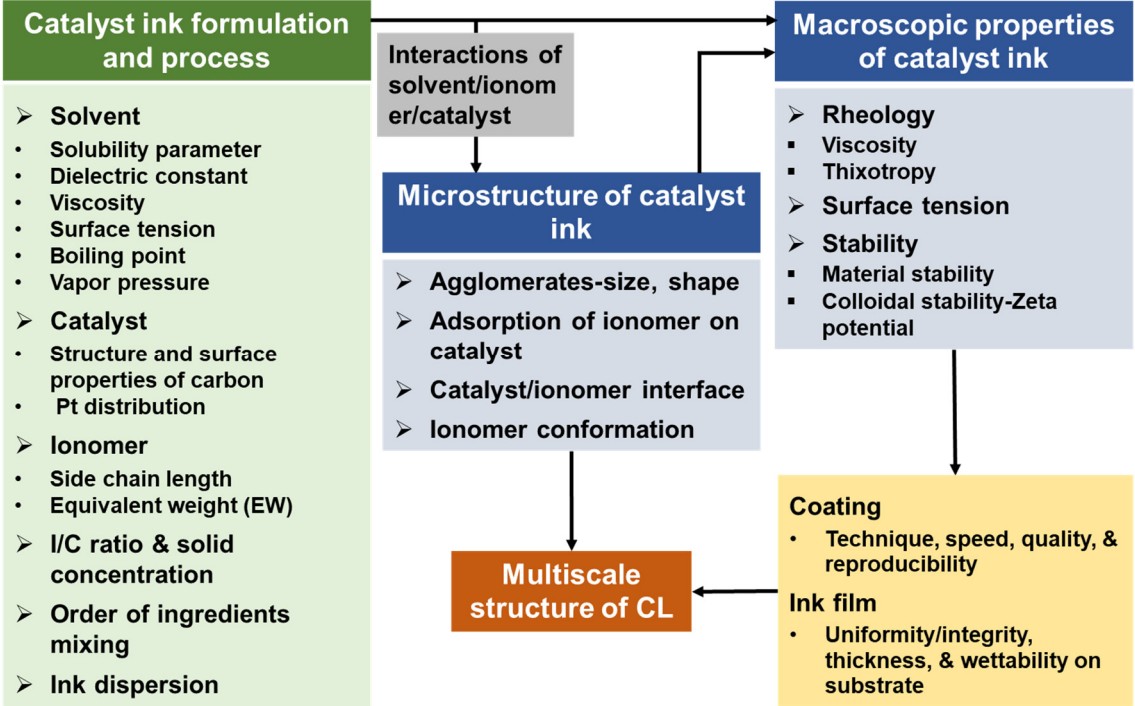

**Figure 4.** The effect of the formulation and preparation processes of catalyst ink on its microstructure and macroscopic properties and further on the fabricated CLs.

To thoroughly investigate the microstructure of the catalyst ink, there are various characterization techniques applied, e.g., Dynamic Light Scattering (DLS), Small Angle Scattering (SAS), [19]F Nuclear Magnetic Resonance ([19]F NMR, always used to investigate the morphology of ionomer in solvent), Transmission Electron Microscope (TEM), Scanning Electron Microscope (SEM), Cryogenic TEM (Cryo-TEM), Cryo-SEM, X-ray computed tomography (X-CT) [26,61–65]. The SAS includes Small Angle Neutron Scattering (SANS) and Small Angle X-ray Scattering (SAXS), as well as ultra-small-, and wide-angle X-ray scattering, contrast variation SANS (CV-SANS). The DLS is often used to measure the hydrodynamic radius of the primary and secondary agglomerates in the catalyst ink. One of its primary drawbacks is that the ink needs to be diluted to avoid excessive scattering and guarantee sufficient accuracy, which would lead to a few discrepancies between the DLS results and the real case due to the different concentration scale. Shukla et al. found that the decrease in the primary and secondary agglomerates size resulting from dilution approaches 20%, through studying the ink with and without dilution by using a recent DLS apparatus [34]. TEM and SEM need to modify the concentration of catalyst ink as well,

i.e., consolidation, and merely characterize the local structure. DLS, TEM, and SEM might not truly investigate the microstructure of catalyst ink. Compared to DLS, TEM, and SEM, SAS and X-CT may be more suitable to characterize the microstructure of catalyst ink since the catalyst ink could be investigated without modification.

### 3.2. Macroscopic Properties of the Catalyst Ink

The macroscopic properties of the catalyst ink, including rheology (viscosity and thixotropy), surface tension, and stability (material stability and colloidal stability), are impacted by the catalyst ink composition or microstructure of catalyst ink. The macroscopic properties of the catalyst ink highly influence the coating process and the deposited ink film, hence the multiscale structure of CL (Figure 4).

(1) Rheology: During the coating process, the ink with the desired viscosity is necessary for high-quality, continuous, and defect-free coating. Some coating techniques, e.g., slot die coating, doctor blade coating, brush coating, or print coating, usually need a relatively viscous ink, often called slurry or paste. A too viscous paste results in poor flowability, leading to a poor distribution of ink on the substrate or weak adhesion of CLs to the membrane [66]. Inkjet printing and spray coating generally need a low viscous ink (e.g., 4.9 mPa s) to avoid blocking the spraying nozzle and decrease the structural defects in the CLs, for example, the ink with a viscosity of 25.3 mPa s [67] may result in the nozzle clogging, making it hard to control the amount of ink sprayed. For spray coating, the viscosity has an influence on the atomization of the catalyst ink. When the viscosity is lower than a certain critical value (e.g., 15 mPa s [68]), increasing the viscosity of ink will decrease the number of large droplets and increase the uniformity of the formed droplets, enhancing spraying quality.

The viscosity of catalyst ink is generally determined by temperature, the viscosity of solvent, the volume fraction of solids in the ink, and shear rate [53].

(i) Temperature, the viscosity of solvent, and the volume fraction of solids in the ink. The viscosity of a liquid always decreases with an increase in temperature. A viscous solvent or an increase in the solid volume in the ink lead to an increase in ink viscosity. Here, the solid refers to the catalyst, ionomer, and their formed agglomerates, as well as the solvent trapped in the loose agglomerates, which is equal to a pseudo-solid phase [52]. For a certain solid content, the ink viscosity usually increases with decreasing the size of agglomerates [28,69]. Furthermore, the viscosity will decrease when the strength of the electrostatic repulsion force between agglomerates increases [53].

(ii) Shear rate: When applying a shear force, e.g., during dispersing or coating of the catalyst ink, the fluid properties change, e.g., viscosity, different from the stationary case, which is commonly due to the change of the fluid microstructure caused by shear force [70]. The rheological behavior of fluid could be studied by measuring the change of viscosity or shear stress ($\tau$) with shear rate ($\gamma$). Fluid can be categorized into Newtonian fluids with shear-independent viscosity and non-Newtonian fluids. The non-Newtonian flow behavior could be further categorized in shear-thickening fluids and shear-thinning fluids. The Newtonian fluid exhibits a linear function between the applied $\tau$ and the rate of strain (or $\gamma$). The non-Newtonian fluid could exhibit such a nonlinear function, called the Ostwald–de Waele model or Power Law model:

$$\tau = \kappa \gamma^n = \left( \kappa \gamma^{n-1} \right) \gamma = \eta \gamma \tag{1}$$

where $\kappa$ is the rheological consistency index, and $\eta$ is the apparent viscosity; the exponent $n$ can be determined if the applied $\tau$ and the resulting $\gamma$ are measured. For a Newtonian fluid, $n = 1$, and the apparent viscosity is independent on $\gamma$. If $n \neq 1$, the fluid is termed non-Newtonian, or pseudo-plastic. The apparent viscosity increases with $\gamma$ when $n > 1$, i.e., shear-thickening fluid, or decreases with the $\gamma$ when $n < 1$, i.e., shear-thinning fluid [71]. The difference between n and 1 determines the degree of shear-thinning or shear-thickening.

The larger the difference, the higher the shear-thinning or shear-thickening behavior. There are several other visco-elastic models to describe shear-thinning flow behavior, e.g., the Carreau-Yasuda-Cross Model [72], or for fluids exhibiting a yield stress, the Herschel-Bulkley Model [73].

For non-Newtonian fluid, the microstructure variation will occur under shear force. For example, the large agglomerates are broken down to small agglomerates by shear force; the initially tangled polymer chains disentangle and stretch under shear force; the initially disordered particles align with flow [71]. The fluids exhibiting these structural variations will show a shear-thinning behavior. The fluids exhibiting the reversed structure variations will show a shear-thickening behavior.

The catalyst ink is a multi-phase fluid system, likely exhibiting non-Newtonian fluid behavior when flowing. The commonly used water and alcohol mixed solvents are Newtonian fluids. When dispersing Nafion ionomer in the solvent with a weight ratio from 0.25 to 5.5 wt.%, the dispersion solution still approaches a Newtonian behavior for shear rates above $10 \, \mathrm{s}^{-1}$ [52,53]. When dispersing Pt/HSC (high surface carbon) in the solvent, the fluid exhibits a shear-thinning behavior. When simultaneously dispersing Pt/HSC and Nafion ionomer in the solvent, i.e., the catalyst ink, it presents a shear-thinning nature [53] (Figure 5). To the best of our knowledge, most catalyst inks usually give a shear-thinning behavior [28,52,53,74]; only a few cases exhibit a nearly Newtonian behavior for shear rate of about $1 \sim 1000 \, \mathrm{s}^{-1}$ [53,75]. Under shear force, the primary or secondary agglomerates in the catalyst ink will align with the flow, or the large secondary agglomerates break down to small secondary agglomerates or primary agglomerates, resulting in a shear-thinning behavior. The degree of shear thinning increases with increasing the concentration of catalyst or ionomer due to the increased degree of agglomeration [53], as shown in Figure 5b,c. Badly dispersed carbon inks with dry agglomerates could exhibit shear thickening behavior at critical shear rates, whereas the overall flow behavior is shear thinning [76,77].

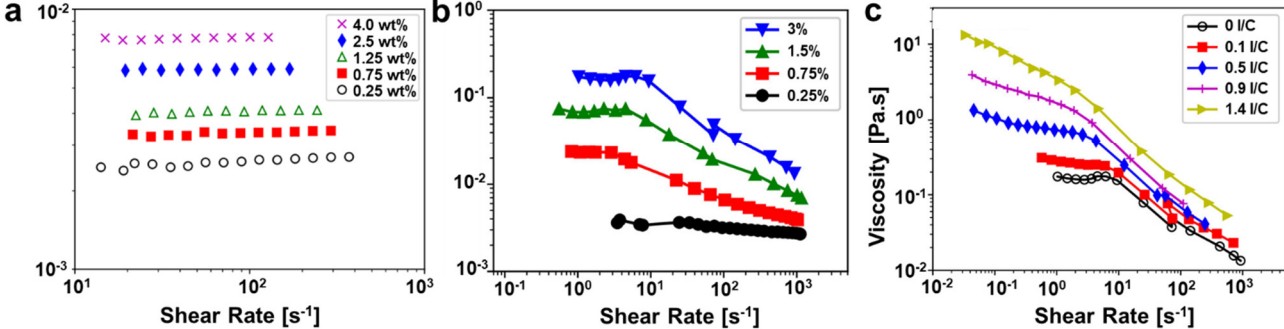

**Figure 5.** Viscosity of Nafion ionomer dispersions (**a**), Pt/HSC dispersions (**b**), Pt/HSC and ionomer dispersions with a fixed 3 wt.% carbon (**c**) in 1-propanol and aqueous solvent mixture (3:1 mass ratio) at different concentrations or I/C ratios (Reprinted with permission from ref. [53]. Copyright (2018) American Chemical Society).

When through a spray nozzle, doctor blade, or brush, the catalyst ink is subjected to the different shear environments, e.g., through a spray nozzle, a high shear environment (the shear rate at the ink film is ca. $10^4$–$10^6 \, \mathrm{s}^{-1}$ in the spraying process [68]), through a brush, a low shear environment exists [35]. Especially, the flow behavior at high shear rates ($>10^4 \, \mathrm{s}^{-1}$) is hardly measurable with rotational rheometers due to edge fracture and spillage. Therefore, modeling the flow behavior could give an indication of the visco-elastic behavior at high shear rates during the coating process [78]. If the fluid approaches a Newtonian behavior, the catalyst ink would suffer nearly no microstructure variation during the coating process. The deposited ink film keeps a similar microstructure to that of the catalyst ink. However, in most cases, the catalyst ink is a shear-thinning fluid. During the coating process, the deposited ink film suffers deformation, showing a different microstructure from the catalyst ink. Such deformation complicates the relation between

the catalyst ink microstructure and the CL structure as well as increase the difficulty in controlling the CL structure. However, the shear-thinning behavior will contribute to the uniformity of the coated CL and controlling the CL thickness during the coating process.

The viscosity and rheology of catalyst ink are sensitive to its microstructure. Therefore, the viscosity and rheology measurements could be used to probe the microstructure of the catalyst ink or investigate the effect of the catalyst ink ingredients and preparation process on its microstructure [52].

When taking the shear force off, the shear-thinning fluid needs to take time to return to the initial viscous state, generally called thixotropy. In addition to visco-elastic behavior, thixotropy is also essential for the coating process, especially for the roll-to-roll coating techniques due to the commonly used ink with a high solid concentration and the fast-coating speed. During the coating process, an appropriate thixotropic response will contribute to the formation of the CL with uniform thickness and flat surface. If the ink viscosity recovery time is too short, it is difficult to obtain a good leveling, forming an uneven surface. However, a long recovery time will cause a hanging phenomenon, making it hard to form an ink film with enough thickness. The microstructure of ink film may not return to the initial ink state before completely drying. The thixotropy of ink could be regulated by varying the dispersion method [28] or adjusting the dielectric constant of solvent or varying I/C ratio [79].

(2)  Surface tension: During the coating process, the surface tension of the ink should be minimized, or it should be at least less than the substrate, to boost the ink's wettability on the substrate and avoid coating failures. Hence, low surface tension will be beneficial to decrease the formation of defects, enhance the adhesion of CLs to the substrates, and optimize the interface between the CL and substrate. For example, to allow adequate wetting during the coating process onto carbon cloth, the surface tension value of ink should be lower than 32.5 mN/m (the surface tension value of carbon cloth) [80]. During the spraying process, the surface tension of the ink could impact the spray quality to some extent by affecting the small droplets [68].

During the coating process, the rheological behavior, including viscosity and thixotropy, determines the catalyst ink's response to shear stress, and surface tension of the catalyst ink play a critical role in the coating process (e.g., coating speed or quality) and the quality of the deposited ink film (e.g., uniformity, integrity, thickness, or wettability on substrate) (Figure 4).

(3)  Stability: Not only the zero-time microstructure and macroscopic properties of a catalyst ink, but also its stability is necessary to be considered to guarantee reproducible coating, which is extremely critical for industrial production. The stability of the catalyst ink includes material stability and colloidal stability. The former includes the oxidation of carbon, ionomer, and organic solvent caused by the catalysis of Pt or dealloying of catalyst during storage, modifying the composition of catalyst ink and as a result, impacting the structure or performance of CL [64,81]. The latter refers to the ability of the agglomerates in the fresh ink to remain dispersion in the solvent. Over time, the well-dispersed agglomerates may collide with each other during random Brownian motion and aggregate together when the collision force is larger than the repulsive force. If the agglomerate size reaches a certain point, sedimentation would take place due to gravity, making the catalyst ink inhomogeneous [34,52].

Zeta potential is commonly used as an indicator of colloidal stability. The zeta potential of particles in the colloidal system is the electric potential at the outside of the stationary layer on the particle surface, i.e., the electric potential at the shear plane of the particle, which could be measured by Electrophoretic Light Scattering (ELS) [63]. Its magnitude (negative or positive) indicates the degree of electrostatic repulsion between neighboring particles with the same charge. In general, the larger its absolute value, the more stable the dispersion. When its absolute value is less than 10 mV, the electrostatic repulsion force between the adjacent particles is too weak to resist attractive forces, e.g., Van De Waals force,

such that the particles tend to aggregate, and the colloid is unstable. When the absolute value of Zeta potential is larger than 30 mV, the electrostatic repulsion force is believed to be strong enough to keep the particles from aggregation and the colloid is stable [82]. However, the ink stability is not only related to zeta potential because it does not cover all the interactions between agglomerates, such as the steric effect caused by polymers, i.e., ionomers [82,83].

## 4. Catalyst Ink: Effect of Its Formulation on Its Microstructure and Macroscopic Properties

To practically apply the advanced catalysts with high activity and stability in the PEM fuel cells and boost the performance and durability of fuel cells, the CL with the well-desired multiscale structure is necessary to be fabricated by optimizing the microstructure and macroscopic properties of the catalyst ink and the CL formation process. The microstructure and macroscopic properties of the catalyst ink are closely related to the catalyst type, ionomer, and solvent, I/C ratio, and the solid content, as well as the underlying interactions between them.

### 4.1. Catalyst and Catalyst Content in the Ink

In this review, the catalysts mainly refer to carbon-supported Pt-based catalysts, excluding non-precious group metal catalysts, as the former is the most promising for commercialization at this time. The structure and surface properties of catalysts and catalyst content influence the microstructure and macroscopic properties of a catalyst ink, e.g., the size of catalyst agglomerates, the ionomer coverage on Pt or carbon, viscosity, rheology, or optimal I/C ratio.

(1)　The surface area and structure of carbon support and the distribution or loading of Pt on carbon [74,84–86]: For Pt/Vulcan, the Vulcan carbon black is a solid carbon support; most Pt nanoparticles are on the external surface of Vulcan. However, for Pt/Ketjen, Ketjen carbon black is a porous carbon support and has a high surface area and abundant internal pore; only ∼50% or even less than 50% of Pt nanoparticles reside on the external surface [87,88]. Compared to Pt/Vulcan, Pt/Ketjen has a low Pt surface density. The ionomer exhibits a preferential interaction with Pt relative to carbon through the side chain [50,89]. Therefore, Pt/Vulcan has a more uniform ionomer coverage than Pt/Ketjen due to higher Pt surface density [53,59,60]. In the catalyst ink, the Pt/Vulcan-related agglomerates can be effectively stabilized by ionomer via the electro-steric mechanism. The viscosity of Pt/Ketjen-based ink is generally higher than that of Pt/Vulcan-based ink [52,53]. However, Yoshimune et al. recently observed that increasing Pt loading on carbon would lead to reduced density and increased thickness of the adsorbed ionomer layer in the water/ethanol-based catalyst ink, due to the hydrophobic interaction between the main chain of ionomer and carbon [90]. Furthermore, the support [88,91] (e.g., nanopore volume [88]) or Pt loading on carbon [92,93] will impact the optimal I/C ratio; certainly, the optimal I/C ratio also depends on the type of ionomer (discussed more in Section 4.2), Pt loading in the electrode [94], or operation conditions (e.g., RH [95]). Moreover, pure carbon instead of Pt/C was usually used in some studies to investigate the effect of the structure and surface properties of carbon on the ink microstructure [74], which is improper since the surface properties of carbon may be modified during depositing Pt [53].

(2)　The amount and charge type of functional groups on carbon [51,85,96]: For example, hydrophobic carbon tends to form large agglomerates in the polar solvent, e.g., water/alcohol mixed solvent, due to hydrophobic interaction [25]; in contrast, increasing the hydrophilicity of carbon through introducing plenty of functional groups tends to form small agglomerates in polar solvents due to an enhanced repulsion force [51]. To improve the dispersion of Pt on carbon, the pristine carbon materials are usually subjected to the oxidation treatment to increase the surface oxygen-containing groups. This makes the carbon surface negatively charged, increasing the columbic repulsive force with the negatively charged ionomer, which is adverse to the adsorp-

tion of ionomer on catalyst [97]. The equilibrium constant of ionomer adsorption on catalyst decreases with increasing the amount of the surface oxygen-containing groups [96]. If introducing -SO$_3$H onto the carbon surface, a similar phenomenon will be incurred [51]. However, if the surface of the carbon in catalyst contains -NHx or N, the interaction between ionomer and catalyst will be enhanced due to the electrostatic attractive interaction, increasing the coverage and uniformity of ionomer on the catalyst.

In brief, the catalyst's structure and surface properties would influence the interaction of catalyst⎮catalyst, catalyst⎮ionomer, or catalyst⎮solvent, which will impact the microstructure and macroscopic properties of the ink made of the catalyst and further the CL structure. Therefore, it is necessary to optimize the ink formulation for the advanced catalysts rather than using the same recipe as the commercial Pt/C catalyst, to improve the CL structure and then enhance the performance of PEM fuel cell with the advanced catalysts. For example, the optimum I/C ratio needs to be determined for various catalysts, i.e., empirically determined optimum Nafion content in the electrode, 30 wt.% [98], is not appropriate for all catalysts [91,99].

(3)    The modification of Pt nanoparticles: Although it is beneficial for proton transport to Pt, the ionomer film covering Pt increases the kinetic polarization loss due to the poisoning of sulfonate groups [100], and the mass transport polarization loss due to increased local gas transport resistance through ionomer layer and the increased interfacial resistance caused by the interaction of ionomer and Pt, especially in the CL with a low Pt loading [101]. To prevent Pt surface from being covered by ionomer, the Pt surface is covered by alkanethiol, acting as a mask, before mixing catalyst and ionomer (Figure 6). In this case, the ionomer will selectively cover the carbon surface. After the CL is fabricated, the absorbed alkanethiol molecules are removed by voltage cycling, releasing the Pt surface. Therefore, there is barely any ionomer molecules covering the Pt surface, enhancing local gas transport [102,103]. However, the effect of the masking and demasking process on the activity of Pt nanoparticles may need attention.

(4)    Catalyst content in the ink: If the catalyst content in the ink is increased, the amount and size of agglomerates increase, and the distance between agglomerates decrease, increasing the viscosity of ink [52,53]. The catalyst content in ink commonly relies on the ink deposition method, e.g., spray coating usually needs a lower catalyst content (the solid concentration is commonly less than 3 wt.%, e.g., ~0.6 wt.% [36], ~1.8 wt.% [104], or 2.5 wt.% [67]) to avoid the blockage of spraying nozzle, while roll-to-roll coating commonly requires a high catalyst content (the solid concentration is commonly larger than 5 wt.%, e.g., 5.76 wt.% [105], or ~4.5–15 wt.% [36], or even 33.76 wt.% for screen printing [80]). However, the ink for spraying cannot be diluted too much, resulting in a low fabrication rate and a waste of solvent to deposit a desired Pt loading. Therefore, the catalyst content should be adjusted for different coating techniques in order to improve the coating quality as well as decreasing the coating time or the fabrication cost of CL.

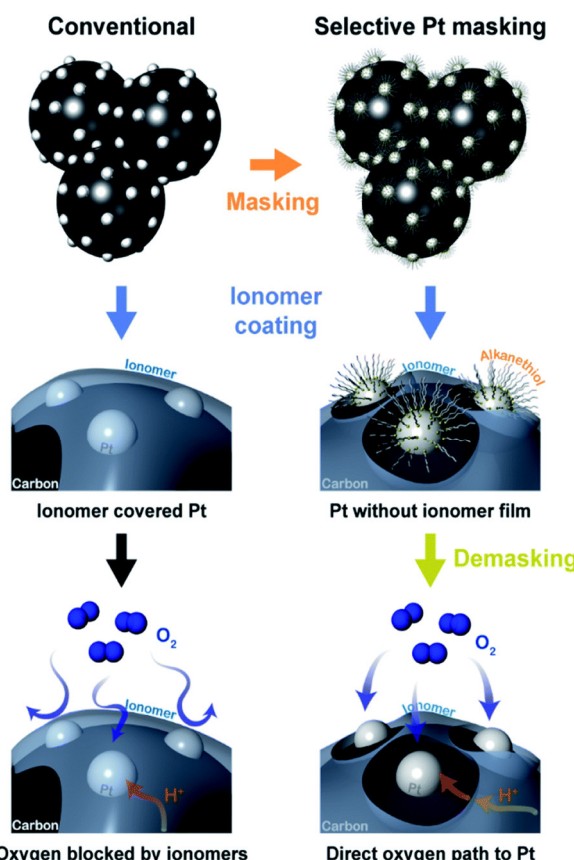

**Figure 6.** Schematics of the distribution of ionomer on carbon and Pt nanoparticles in the conventional case (**left**), the selective distribution of ionomer on only carbon surface through masking Pt nanoparticles (**right**), as well as the oxygen/proton transport in both cases. (Reprinted from Ref. [103] with permission from the Royal Society of Chemistry).

### 4.2. Ionomer and I/C Ratio

The ionomer is a critical constituent in CL, which contributes to (i) increasing proton conductivity from the active sites to the membrane, (ii) extending the TPB, (iii) holding catalyst particles together, and (iv) keeping the mechanical integrity of the CL during fabrication and operation. Perfluorinated sulfonic acid (PFSA) random copolymer is one class of the most commonly used ionomers in CLs. PFSA ionomers consist of a hydrophobic polytetrafluoroethylene main chain and hydrophilic side chains terminated with a sulfonic acid group as shown in Figure 7. Such architecture would induce the main chains to aggregate with side chains surrounding in the polar solvent. The physicochemical properties of PFSA ionomer could be modified by varying the side chain length or equivalent weight (EW, ionomer weight per mole of sulfonic acid groups, expressed as grams per mole) [106,107], which influences the microstructure and macroscopic properties of the catalyst ink by impacting the interaction with catalyst or solvent. Furthermore, the I/C ratio has a vital effect on the microstructure and macroscopic properties of the catalyst ink and further significantly impacts the CL structure.

(1) The type of ionomer: According to the side chain length, the ionomers can be categorized into long-side chain (LSC) ionomer, e.g., Nafion produced by DuPont or perfluoroimide acid (PFIA) ionomer produced by 3M, and short-side chain (SSC) ionomer, e.g., PFSA ionomer produced by 3M and Aquivion ionomer produced by Solvay-Solexis (Figure 7). The commonly used commercial ionomer dispersion solutions and their related parameters are summarized in Table 1. According to the EW, the ionomers can be categorized into low, middle, and high EW ionomer. 3M PFIA

ionomer has multiple acid groups per side chain, while the others only have a single acid group at the end of each side chain. Hence, the 3M PFIA ionomer possesses a high ion exchange capacity (IEC). Since it is a newly developed ionomer, there are only a few studies exploring the properties and application of 3M PEIA ionomer [108,109]; therefore, the LSC ionomers mainly refer to Nafion ionomer in this review. The EW of ionomer depends on the number (m) of -$CF_2$- repeat units in the main chain and the molecular weight of the side chain ($MW_{sc}$), i.e., EW = 50 m + $MW_{sc}$. For a specific EW, compared to a LSC ionomer, a SSC ionomer has a larger m value, i.e., a larger main chain fraction, resulting in higher crystallinity and thus higher glass transition temperature and enhanced thermal stability [110]. Hence, the SSC ionomer can be prepared with a low EW on the premise of the sufficient residual crystallinity (i.e., adequate stability upon water uptake rather than dissolving or becoming gelatinous), leading to higher proton conductivity. EW and the side chain length are two important parameters influencing the physicochemical properties of ionomer. Recently, Ramaswamy et al. [111] found that EW is the major factor influencing the performance of PEM fuel cell at the high current region rather than side chain length through systematically analyzing the effect of the side chain length and EW of ionomers on the transport resistances of CL. According to the results Young-Chul Park et al. reported, the side chain length may determine the performance of PEM fuel cells at the low current region [107]. The better continuity and uniformity of SSC ionomer on Pt and carbon increase Pt utilization [107].

**Figure 7.** The chemical structures of 3M PFIA ionomer, Nafion ionomer, 3M PFSA ionomer, and Aquivion ionomer [108,112].

In addition to the enhanced thermal stability and proton conductivity, SSC ionomers commonly exhibit several advantages compared to Nafion ionomer, less sensitive to the change of solvent (e.g., the water content in dipropylene glycol (DPG) and water mixed solvents [112]), as well as increased main chain mobility, solubility, and dispersibility in the solvent, resulting in relatively weaker clustering tendency and smaller agglomerate size [113]. For example, the SSC ionomer with low EW (IEC = 1.43 meq/g) is better dispersed in polar solvents (e.g., a large ratio of water to alcohol) than the LSC ionomer with high EW (ion-exchange capacity 0.99 meq/g), due to the stronger interaction between sulfonic acid groups and water than alcohol [107]. Therefore, LSC and SSC ionomers might need different solvents or mixed solvents to achieve the optimal dispersion and conformation. In brief, the CLs containing SSC ionomer usually show improved and con-

nected secondary pores, uniform and continuous ionomer coverage on the catalyst, and enhanced ionomer distribution, leading to higher Pt utilization efficiency and improved mass transport [106,107,113,114]. The corresponding PEM fuel cells usually exhibit increased performance, particularly at mass transport polarization region, enhanced durability, and less sensitivity to operation conditions, e.g., RH, operation pressure, or temperature (especially under high operation temperature and low or medium RH due to the effective water trapping ability of the SSC ionomer that usually possesses a low EW) [106,107,113–116].

(2) I/C ratio: If adding a small amount of ionomer in the catalyst ink, i.e., a low I/C ratio, the ionomer molecules adsorb on the catalyst agglomerates, enhancing the columbic repulsion force and steric hindrance of inter-agglomerates [42,53,74], inhibiting aggregation, to some extent similar to a stabilizing agent, decreasing the agglomerate size, and thus improving the colloidal stability [53,54]. When the I/C ratio reaches a certain value, increasing the content of ionomer in the catalyst ink further will not contribute to the size of agglomerates or increase the colloidal stability [34]. It has been reported that the ionomer adsorption on carbon or catalyst follows the Langmuir isotherm for both SSC and LSC ionomer when the ionomer concentration is less than a certain value (at least 1.0 I/C [54]), i.e., there is an adsorption plateau, where the absorbed ionomer content barely increases with increasing I/C ratio [27,54,74]. The amount of the absorbed ionomer on catalyst commonly depends on the surface properties of the catalyst [117], EW of ionomer [27], or the dispersion medium [118]. Moreover, I/C ratio impacts the viscosity of catalyst ink; however, the influencing rule alters with the type of catalyst. For example, in the case of Pt/Vulcan, the ink viscosity slightly decreases (I/C ratio < 0.2) and then increases with the increase in I/C ratio, whereas in the case of Pt/Kejten, the ink viscosity increases over I/C ratio because ionomer flocculates the agglomerates [53].

The I/C ratio in the catalyst ink is a critical parameter impacting the performance and durability of PEM fuel cells, which generally needs to be optimized for different cases to increase Pt utilization and minimize the mass transport resistances. If the I/C ratio is too low to form a continuous network for effective proton transport, the accessibility of Pt to proton is insufficient, decreasing TPB. If the I/C ratio is too high to block the pores of the CL, the blocked pores will hinder the gas and water transport, causing increased local and bulk gas transport resistance or flooding [119]. According to the measured results of CV-SANS, the non-absorbed ionomer exists in the catalyst ink, although the I/C ratio is as low as 0.25, and increases with an increasing I/C ratio; the amount of the absorbed ionomer increases from 0.25 to 0.50 and then is almost constant up to 1.00 I/C (Pt/Vulcan and Nafion D2020 ionomer dispersion) [54]. Andersen et al. [98] found that by using 57 wt.% Pt/C and Nafion ionomer to fabricate the CL, the optimum Nafion content is 30 wt.%. In this case, the I/C weight ratio is near 0.7. When the I/C ratio is less than 0.7, the adsorption of ionomer on catalyst might have been saturated; only the amount of non-absorbed ionomer in catalyst ink increases with the I/C ratio increasing to 0.7. Consequently, in addition to the adsorbed ionomer, the amount, size, and configuration of non-absorbed ionomer agglomerates, which vary with the I/C ratio, will determine the performance of the cell by impacting the continuity of the proton transport network, porosity, pore size, and size distribution, especially the secondary pores. Furthermore, the non-adsorbed ionomer has an influence on the ink viscosity, which increases with increasing I/C ratio or the amount of non-adsorbed ionomer [54]. The adsorbed ionomer and the amount, size, and configuration of non-absorbed ionomer and thus optimum I/C ratio are sensitive to the catalyst, ionomer, and solvent [23,91,120]. For example, it was observed that the optimal ionomer content in the CL is dependent on the catalyst or the EW of the ionomer [120]. When the other conditions are fixed, the optimal I/C ratio for the 3M PFSA ionomer (EW 725, 25 wt.% in CL) is lower than that of Nafion ionomer (EW 1100, 30 wt.% in CL) due to the more uniform distribution in the CL and higher proton conductivity [113].

**Table 1.** The commonly used commercial ionomer dispersion solutions and their related parameters [109,121–124].

| Ionomer Dispersion | | Content (wt.%) | Solvent | EW (g/mol) | IEC (meq/g) | Viscosity (mP s, 25 °C) |
|---|---|---|---|---|---|---|
| Nafion PFSA | D-520 | 5–5.4 | Water (45 wt.%), 1-propanol (48 wt.%) | 890–970 | 1.03–1.12 | 10–40 (40 s$^{-1}$) |
| | D-1020 | 10–12 | Water (87–90 wt.%) | | | 2–10 |
| | D-2020 | 20–22 | Water (34 wt.%), 1-propanol (44 wt.%) | | | 50–500 |
| | D-521 | 5–5.4 | Water (45 wt.%), 1-propanol (48 wt.%) | 970–1050 | 0.95–1.03 | 10–40 |
| | D-1021 | 10–12 | Water (87–90 wt.%) | | | 2–10 |
| | D-2021 | 20–22 | Water (34 wt.%), 1-propanol (44 wt.%) | | | 50–500 |
| 3M PFSA | 725EW | 15 | 1-propanol/water 60/40 (m/m) | 725 | 1.38 | 50–300(20 °C, 1 s$^{-1}$) |
| | 800EW | 20 | | 800 | 1.25 | |
| 3M PFIA | | - | Powder | 620 | 1.61 | — |
| Aquivion PFSA | D72–25BS | 25 | 99% water, free of ethers % | 700–740 | 1.35–1.43 | <25 |
| | D79–25BS | 25 | | 790 ± 20 | 1.23–1.30 | <25 |
| | D83–24B | 24 | >99% water | 810–850 | 1.17–1.23 | 5–15 |
| | D98–25BS | 25 | 99% water, free of ethers % | 940–1020 | 0.98–1.06 | <25 |

*4.3. Solvent*

To optimize the microstructure and macroscopic properties of the catalyst ink and hence the multiscale structure of the CLs made of the ink, there are two common pathways, i.e., the chemical route and physical route. The former is dependent on the chemical methods to change or modify the physicochemical properties of catalyst or ionomer, e.g., the distribution of Pt on carbon, functional groups on the carbon surface, or the side chain of ionomer, as described in Sections 4.1 and 4.2. The latter is based on the physical methods to change the dispersion characteristics of catalyst and ionomer or their interface through varying the dispersion medium [22,125], the order of ingredients mixing, or dispersion process (the latter two are discussed in Section 5). Although solvent evaporates completely and is not present in the CLs, as the dispersion medium, its physical properties exert a significant impact on the microstructure and macroscopic properties of the ink and hence the CL structure. This mainly occurs through its influence on the interactions between ionomer and catalyst, the dispersion of ionomer and catalyst, the interaction between the ink film and substrate, coating process, or drying process [23,26,33,35,42,59,126–128]. The physical properties of a solvent include dielectric constant ($\varepsilon$), solubility parameter ($\delta$), viscosity, boiling point, vapor pressure, and surface tension (Figure 8). The solvents commonly used to prepare the catalyst ink and the solvents that have been used to prepare the catalyst ink in the literature are summarized in Table 2 with their physical properties.

The dielectric constant of a solvent is related to its polarity and reflects the ability to separate opposite charges. For example, water is a polar solvent and has a high dielectric constant value (80), in which NaCl can be dissociated to Na$^+$ and Cl$^-$; hexane is a non-polar solvent and has a small value (1.9).

The solubility parameters provide a numerical estimate of the interaction degree between materials and can be used to indicate the solubility of a polymer in a solvent. Two materials with similar solubility parameters are miscible, following the 'like dissolves like' law. For the time being, there are two kinds of solubility parameters usually used, i.e., Hildebrand solubility parameter and Hansen solubility parameter. Hildebrand solubility parameter ($\delta$) was proposed by Joel H. Hildebrand in 1936 [129], and defined as the square root of the cohesive energy density:

$$\delta = \sqrt{\frac{E_{coh}}{V_m}} = \sqrt{\frac{\Delta H_v - RT}{V_m}} \qquad (2)$$

where $E_{coh}$ is cohesive energy, which can be derived from vaporization heat $\Delta H_v$, $V_m$ is molar volume of a solvent, $R$ is gas constant, and $T$ is temperature. The units of solubility

parameters include $(cal/cm^3)^{1/2}$ (old) and $MPa^{1/2}$ (commonly used now), and 1 $MPa^{1/2}$ is equal to 2.0455 $(cal/cm^3)^{1/2}$ [129].

Hildebrand solubility parameter describes the global interaction, limiting its application, e.g., unsuitable for the case with polar or H-bonding interactions. In 1966, Charles M. Hansen divided the total Hildebrand solubility parameter value into three parameters by taking the effects of dispersion force, polar force, and H-bonding into account, i.e., $\delta_D$, $\delta_P$, and $\delta_H$ [129]. The relation of Hildebrand solubility parameter ($\delta$) and Hansen solubility parameter ($\delta_{total}$) can be described via Equation (3).

$$\delta = \delta_{total} = \sqrt{\delta_D{}^2 + \delta_P{}^2 + \delta_H{}^2} \tag{3}$$

The materials with similar values of $\delta_D$, $\delta_P$, and $\delta_H$ are miscible [129]. Compared to Hildebrand solubility parameter, Hansen solubility parameter is a better indication of the interaction degree between polymer and solvent or predicting the polymer solubility in a solvent.

In addition to pure solvents, the mixtures of two or more miscible solvents, e.g., the commonly used water/alcohol (1-propanol, 2-propanol, or ethanol) mixed solvents, are often used to disperse catalyst and ionomer as well. The physical properties of the dispersion medium will be easily modified by varying the ratio of the mixed solvents. The dielectric constant of the mixed solvents ($\varepsilon_{mix}$) can be calculated via Equations (4) and (5) [26,130].

$$\rho_{mix} = \sum x_i \varepsilon_i{}^{1/2} M_i \tag{4}$$

$$\varepsilon_{mix} = \left( \frac{\rho_{mix}}{\sum x_i M_i} \right)^2 \tag{5}$$

where $\rho_{mix}$ is the molar dielectric polarization of the mixed solvent, $x_i$, $\varepsilon_i$, and $M_i$ are the molar fraction, dielectric constant, and molar mass of solvent $i$, respectively.

Hildebrand solubility parameter of the mixed solvent ($\delta_{mix}$) can be calculated via Equation (6) [26,130].

$$\delta_{mix} = \sqrt{\frac{\sum x_i E_{coh,i}}{\sum x_i V_{m,i}}} \tag{6}$$

where $E_{coh,i}$ and $V_{m,i}$ are the cohesive energy and molar volume of solvent $i$, respectively.

Hansen solubility parameter of the mixed solvent ($\delta_{total,mix}$) can be calculated via Equations (7) and (8) [130].

$$\delta_{j,mix} = \sum \varphi_i \delta_{j,i} \qquad (j = D, \, P, \, H) \tag{7}$$

$$\delta_{total,mix} = \sqrt{\delta_{D,mix}{}^2 + \delta_{P,mix}{}^2 + \delta_{H,mix}{}^2} \tag{8}$$

where $\delta_{j,i}$ is $\delta_D$, $\delta_P$, or $\delta_H$ of solvent $i$, $\delta_{j,\,mix}$ is $\delta_D$, $\delta_P$, or $\delta_H$ of the mixed solvent.

If mixing the solvents with different surface tensions, the surface tension of the mixed solvent is commonly close to the lowest one, as the solvent molecules with low surface tension tend to accumulate at the surface of the mixed solvent. For example, the surface tension value of diacetone alcohol (DAA) is 31 mN/m, lower than that of water (72 mN/m); if mixing DAA into water or even adding only 5 wt.% DAA, the surface tension significantly declines and approaches the value of DAA since DAA molecules favor the solvent/air interface [130]. Hereafter, pure solvents and mixed solvents are collectively called solvents unless otherwise specified.

The recent development on the effect of the solvent physical properties on the microstructure and macroscopic properties of the catalyst ink by impacting the interactions between catalyst, ionomer, and solvent is summarized here.

(1)  Dielectric constant and solubility parameter: The dielectric constant and solubility parameters of a solvent primarily influence the dispersion of catalyst and ionomer or the

conformation of ionomer (absorbed ionomer and non-absorbed ionomer) [31,131,132]. As the catalyst agglomerates are commonly covered by ionomer in the catalyst ink, solvent affects the dispersion or dispersion stability of catalyst likely through impacting the ionomer coverage on catalyst or the conformation of the adsorbed ionomer [34]. Therefore, it should be mainly through impacting the interaction between solvent and ionomer that the solvent influences the microstructure and macroscopic properties of the ink [127]. Consequently, the studies on the dielectric constant and solubility parameter of solvent primarily focus on the dispersion and conformation of ionomer.

Water or water/1-propanol mixed solvent, commonly used as the solvent of the commercial ionomer dispersion solutions as shown in Table 1, are not the good solvent, or the ratio of water and 1-propanol is not optimal. Therefore, to improve the dispersion and conformation of ionomer in catalyst ink, the solvent often needs to be elaborately selected. As described in Section 4.2, the ionomer is composed of main chain and side chains. The main chain is hydrophobic and nonpolar. The side chain is hydrophilic, polar and the terminal $-SO_3H$ group can be dissociated into $-SO_3^-$ and $H^+$ in the protic solvent, e.g., water. It has been reported that Nafion possesses dual solubility parameters, i.e., $\delta 1 = 9.7$ $(cal/cm^3)^{1/2}$ for main chain and $\delta 2 = 17.3$ $(cal/cm^3)^{1/2}$ for side chain [133]. Since the main chain and side chain are incompatible, the solvents with different dielectric constant and solubility parameter have different compatibility with the main chain and side chain, resulting in the ionomer agglomerates with different morphology, size, and conformation [26,32,134].

When dispersed in the various solvents, Nafion agglomerates with different morphology or size have been observed according to the polarization losses of the fabricated CLs, TEM, DLS, SANS, SAXS, $^{19}F$ NMR, or molecular dynamic simulation. For example, the Nafion morphology may form a solution ($\varepsilon > 10$), a colloid ($3 < \varepsilon < 10$), or a precipitate ($\varepsilon < 3$) in some organic solvents with different dielectric constants [135]. Such morphology variation with a dielectric constant is unsuitable for all cases because the dielectric constant is commonly not the single factor involved. The Nafion ionomer is likely to self-assemble into a variety of morphologies with different sizes in different dispersion media (Figure 8b), e.g., cylindrical agglomerates with a diameter of 2–5 nm (in glycerol, ethylene glycol (EG) or water/1-propanol mixed solvents with different 1-propanol fractions) [128,136,137], less-defined and highly solvated agglomerates with the diameter of >200 nm (in water/2-propanol mixed solvents with different 2-propanol fractions) [128], random-coil agglomerates with a radii of gyration of ~4.1 nm similar to a solution behavior (in N-methylpyrrolidone) [128], or secondary agglomerates due to the hydrogen bond between $-SO_3^-$ and $H_3O^+$ (i.e., $SO_3^-\cdots H_3O^+\cdots-O_3S^-$, in water/1-propanol with 20 wt.% 1-propanol) [26]. The morphology and size of the ionomer agglomerates would affect the ionomer coverage on the catalyst agglomerates or the porosity of CLs. It was believed that the colloid ionomer with fine dispersion or specific size adsorbs on catalyst agglomerates, resulting in the formation of the slightly larger agglomerates, which improves the CL microstructure and enhances the porosity and then the gas transport, further the fuel cell performance [23,25,132,138,139]. In brief, although the relation between both is still unclear, the physical properties of solvent have a major effect on the morphology and size of the dispersed ionomer.

In addition to the morphology and size, the conformation of the ionomer in ink also depends on the dielectric constant and solubility parameter of solvents due to the different affinity of the main chain and side chain with solvent molecules. Due to the high affinity with water molecules, the side chains of Nafion tend to be extended out in the water-rich solvent mixtures, which would increase the coulomb repulsion force between Nafion agglomerates, or between catalyst–Nafion agglomerates, or between both, enhancing the colloidal stability of the ink, whereas the main chains tend to aggregate inside due to the low affinity with the polar water molecules [32]. If the solvent is extremely incompatible with the main chain, e.g., pure water, the Nafion will be poorly dispersed, forming large ionomer agglomerates and resulting in weak interaction with catalyst agglomerates, even

losing the binder function and making it difficult to fabricate a complete CL [49]. If the solvent has excellent compatibility with the main chain, e.g., dipropylene glycol [23], the main chains tend to be extended out, and the Nafion ionomer commonly appears to be well dispersed, e.g., the decreased ionomer agglomerate size or approaching solution behavior [32,128]. Furthermore, the ionomer conformation will influence the interface between ionomer and catalyst, including the coverage of ionomer on carbon or Pt, the thickness of ionomer film, and the interaction between ionomer and Pt [22,60]. These would affect the local gas transport resistance through the ionomer layer or the interfacial resistance between the ionomer film and Pt [60].

To date, the studies on the effect of solvents on the dispersion and conformation of ionomer mainly focus on Nafion ionomer. However, the results derived from Nafion might not be applicable to the SSC ionomer in some cases, due to the different distance between the sulfonic acid groups and main chain, which likely results in different conformational orientations in the similar dispersion medium. Recently, Hoffmann et al. studied the effect of Hansen solubility parameters of water/diacetone alcohol mixed solvents on the dispersion and conformation of SSC ionomer (Aquivion D79-25BS). It was found that the Aquivion ionomer always forms solutions in all water/diacetone alcohol mixed solvents, and $\delta_H$ is responsible for the varied affinity between ionomer and solvent with different diacetone alcohol fractions, and the mobility of the main chain declines with increasing diacetone alcohol fraction, different from the case for Nafion [130].

Furthermore, it is necessary to specially summarize the impact of the water/alcohol (methanol, ethanol, 1-propanol, or 2-propanol) mixed solvents most commonly used. The morphology, size, and conformation of the ionomer agglomerates dispersed will differ with the water fraction or the type of alcohol, altering the CLs' structure and their performance [25,30,32,49,57,131,137,140]. While 1-propanol and 2-propanol have similar physical properties, the corresponding inks possess different microstructure or macroscopic properties likely due to the different symmetry of polarity, i.e., 1-propanol—non-symmetric polarity and 2-propanol—symmetric polarity. In the inks containing water/propanol mixed solvents with the same ratio, the ink containing water/1-propanol has smaller agglomerates than the case of water/2-propanol [57]. Furthermore, the ink containing 2-propanol exhibits a stronger shear-thinning behavior and a higher viscosity than that containing 1-propanol [57]. It was reported that cracks [57,58,141] or larger ionomer patches [49] are more likely formed in the CLs made of the ink containing water/2-propanol.

(2)  Viscosity: The solvent viscosity has an effect on the ink viscosity and stability, coating process, and drying process [42,126]. The ink containing a viscous solvent is commonly more stable than that containing a thin solvent due to the decreased Brownian motion of agglomerates and thus decreasing their collision chance [42]. The ink for brush coating, blade coating, or print coating usually needs the relatively viscous solvent, while the ink for spray coating needs the solvent with low viscosity. Moreover, the solvent viscosity may impact the sedimentation and stacking of agglomerates during solvent evaporation due to the different fluidity in the solvents with various viscosity and then affect the pore size and pore volume of the CL. In the ink containing a thin solvent, agglomerates have a tendency to be packed densely, forming many small pores in the CLs, whereas in the ink containing a viscous solvent, agglomerates tend to be packed loosely due to the low fluidity, forming many pores but less small pores in CLs [126].

(3)  Boiling point and vapor pressure: The higher the boiling point or the lower the vapor pressure, the smaller the solvent evaporation rate at the special temperature or RH, influencing the coating and drying processes. A fast evaporation rate, e.g., methanol, will make the ink unstable during coating process in which the catalyst ink is exposed to air, e.g., brush coating, screen printing, or gravure and flexographic printing. Fast evaporation could also lead to surface cracking in the dried CLs [42]. A low solvent evaporation rate could increase the ink stability during the coating process, increasing the drying time in return [66]. Sometimes it is difficult to completely

remove the solvents with a high boiling point, e.g., glycerol [142], or N-methyl-2-pyrrolidone [143]. The residual solvent will block the pores of CLs, decreasing the gas and water transport. Therefore, it is necessary to select the solvents with an appropriate window of boiling point or vapor pressure to guarantee the sufficient stability of ink during the manufacturing process and the complete evaporation of the solvent.

(4) Surface tension: The surface tension of solvent determines the surface tension of the ink to some extent. However, the addition of catalyst and ionomer will change the surface tension [33]. Therefore, the surface tension of the ink is determined by solvent, ionomer, and catalyst.

Compared to modifying catalyst or ionomer, controlling the physical properties of solvent is an easier pathway and could provide much more opportunity and possibility to tailor the microstructure and macroscopic properties of the ink. The physical properties of the dispersion medium could be finely altered by varying the type of solvent or through using the mixed solvent and varying their ratio. The effect of solvent on the microstructure and macroscopic properties of catalyst ink and then the CL structure commonly results from the cooperative effect of various physical parameters. Therefore, it is improper to use only one physical parameter of solvent as the indicator, such as the dielectric constant. In some cases, one or two physical parameters of solvent may be directly related to the performance of the PEM fuel cell. For example, the power density of PEM fuel cells increases with an increase in the dielectric constant and Hildebrand solubility parameter of water/1-propanol (or methanol) mixed solvents [131].

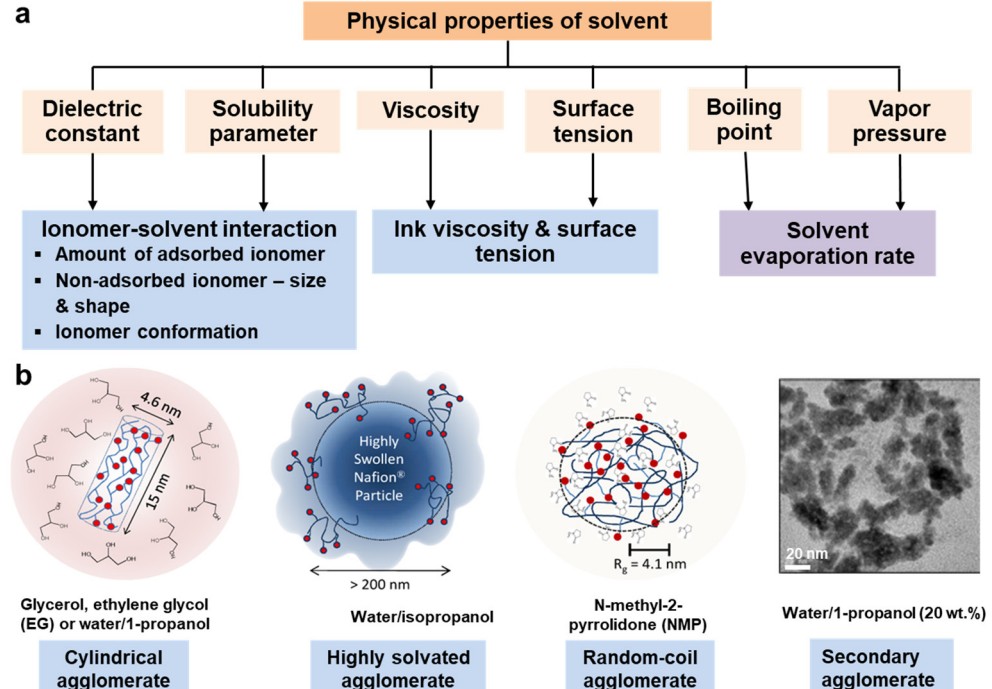

**Figure 8.** (**a**) The physical properties of solvent and their effects on the microstructure and macroscopic properties of catalyst ink, or evaporation rate; (**b**) the size and morphologies of Nafion ionomer in different solvents (Adapted with permission from [128], Copyright (2012) American Chemical Society; Reprinted from [26], Copyright (2013), with permission from Elsevier).

**Table 2.** The dielectric constant ($\varepsilon$), viscosity, boiling point, surface tension and solubility parameters ($\delta$ and $\delta_{total}$) of the solvents commonly used in catalyst ink and the solvents that have been used in the literature to prepare catalyst ink (arranged by boiling point), and the solubility parameters of several ionomers.

| Solvent | Structure | $\varepsilon$ @ 20 °C [144] | Viscosity (cp or mPa s) @ 25 °C [144] | Boiling Point (°C) | Surface Tension (mN/m) @ 25 °C [144] | $\delta$ and $\delta_{total}$ * (MPa$^{0.5}$) @ 25 °C [145] | $\delta_D$ ** [145] | $\delta_P$ ** [145] | $\delta_H$ ** [145] |
|---|---|---|---|---|---|---|---|---|---|
| Diethyl ether | $(C_2H_5)_2O$ | 4.2666 | 0.224 | 34.6 | 16.5 | 15.8 | 14.5 | 2.9 | 5.1 |
| n-Pentane | $CH_3(CH_2)_3CH_3$ | 1.8371 | 0.224 | 36 | 15.45 | 14.5 | 14.5 | 0 | 0 |
| Dichloromethane | $CH_2Cl_2$ | 8.93 (25 °C) | 0.413 | 39.6 | 27.20 | 20.2 | 18.2 | 6.3 | 6.1 |
| Acetone | $CH_3COCH_3$ | 21.01 | 0.306 | 56 | 22.71 | 19.9 | 15.5 | 10.4 | 7.0 |
| Chloroform | $HCCl_3$ | 4.8069 | 0.537 | 61 | 26.65 | 19.0 | 17.8 | 3.1 | 5.7 |
| Methanol | $CH_3OH$ | 33.0 | 0.544 | 65 | 22.17 | 29.6 | 15.1 | 12.3 | 22.3 |
| Tetrahydrofuran (THF) | $(CH_2)_4O$ | 7.52 (22 °C) | 0.456 | 66 | 26.4 [146] | 19.4 | 16.8 | 5.7 | 8.0 |
| Hexane | $CH_3(CH_2)_4CH_3$ | 1.8865 | 0.3 | 69 | 17.88 | 14.9 | 14.9 | 0 | 0 |
| Ethyl acetate | $CH_3-COO-CH_2-CH_3$ | 6.0814 | 0.423 | 77 | 23.39 | 18.2 | 15.8 | 5.3 | 7.2 |
| Ethanol | $CH_3CH_3OH$ | 25.3 | 1.074 | 79 | 21.91 | 26.5 | 15.8 | 8.8 | 19.4 |
| Cyclohexane | $C_6H_{12}$ | 2.0243 | 0.625 | 81 | 24.42 | 16.8 | 16.8 | 0 | 0.2 |
| 2-propanol (IPA) | $CH_3-CHOH-CH_2$ | 20.18 | 2.04 | 82 | 20.92 | 23.6 | 15.8 | 6.1 | 16.4 |
| Triethylamine | $N(CH_2CH_3)_3$ | 2.418 | 0.347 | 90 | 20.22 | 18.8 | 17.8 | 0.4 | 1.0 |
| 1-Propanol (NPA) | $CH_3CH_2CH_3OH$ | 20.8 | 1.945 | 97 | 23.37 | 24.6 | 16.0 | 6.8 | 17.4 |
| n-Heptane | $CH_3(CH_2)_5CH_3$ | 1.9209 | 0.387 | 98 | 19.73 | 15.3 | 15.3 | 0 | 0 |
| 2-Butanol | $CH_3CH_2CHOHCH_3$ | 17.26 | 3.10 | 99 | 24.13 | 23.2 | 16 | 5.7 | 15.8 |
| Water | $H_2O$ | 80.1 | 0.89 | 100 | 72.06 | 47.8 | 15.6 | 16.0 | 42.3 |
| Formic acid | $HCOOH$ | 51.1 (25 °C) | 1.607 | 101 | 37.13 | 24.9 | 14.3 | 11.9 | 16.6 |
| Toluene | $C_6H_5-CH_3$ | 2.38 (23 °C) | 0.56 | 111 | 27.91 | 18.2 | 18.0 | 1.4 | 2.0 |
| Pyridine | $C_5H_5N$ | 13.26 | 0.879 | 115 | 36.56 | 21.8 | 19.0 | 8.8 | 5.9 |
| Acetic acid | $CH_3COOH$ | 6.2 | 1.056 | 118 | 27.10 | 21.4 | 14.5 | 8.0 | 13.5 |
| 1-Butanol | $CH_3CH_2CH_2CH_2OH$ | 17.84 | 2.54 | 118 | 24.13 | 23.2 | 16.0 | 5.7 | 15.8 |
| n-Octane | $CH_3(CH_2)_6CH_3$ | 1.948 | 0.508 | 125 | 21.17 | 15.5 | 15.5 | 0 | 0 |

**Table 2.** *Cont.*

| Solvent | Structure | $\varepsilon$ @ 20 °C [144] | Viscosity (cp or mPa s) @ 25 °C [144] | Boiling Point (°C) | Surface Tension (mN/m) @ 25 °C [144] | $\delta$ and $\delta_{total}$ * (MPa$^{0.5}$) @ 25 °C [145] | $\delta_D$ ** [145] | $\delta_P$ ** [145] | $\delta_H$ ** [145] |
|---|---|---|---|---|---|---|---|---|---|
| n-Butyl acetate | $CH_3\text{-}CO\text{-}O\text{-}(CH_2)_3CH_3$ | 5.07 | 0.685 | 126 | 24.88 | 17.4 | 15.8 | 3.7 | 6.3 |
| Isoamyl alcohol | $(H_3C)_2CHCH_2CH_2OH$ | 15.63 | 3.69 | 131 | 23.73 | 21.3 | 15.8 | 5.2 | 13.3 |
| p-Xylene | $C_6H_4(CH_3)_2$ | 2.2735 | 0.603 | 138 | 27.80 | 17.9 | 17.6 | 1.0 | 3.1 |
| N,N-dimethylformamide (DMF) | $(CH_3)_2NCHO$ | 38.25 | 0.794 | 153 | 35.74 | 24.9 | 17.4 | 13.7 | 11.3 |
| Cyclohexanol | $HOCH(CH_2)_5$ | 16.4 | 57.5 | 162 | 33.25 | 22.4 | 17.4 | 4.1 | 13.5 |
| Diacetone alcohol (DAA) | $CH_3C(O)CH_2C(OH)(CH_3)_2$ | 18.2 | 2.8 | 168 | 31 [147] | 20.8 | 15.8 | 8.2 | 10.8 |
| Aniline | $C_6H_5\text{-}NH_2$ | 7.06 | 3.85 | 184 | 42.12 | 22.6 | 19.4 | 5.1 | 10.2 |
| 1,2-Propylene glycol (PG) | $HOCH_2\text{-}CHOH\text{-}CH_3$ | 27.5 (30 °C) | 40.4 | 187 | 35.99 | 30.2 | 16.8 | 9.4 | 23.3 |
| Dimethyl sulfoxide (DMSO) | $(CH_3)_2SO$ | 47.24 | 1.987 | 189 | 42.92 | 26.7 | 18.4 | 16.4 | 10.2 |
| Octanol | $CH_3(CH_2)_7OH$ | 10.3 | 7.29 | 195 | 26.96 | 20.6 | 16.0 | 5.0 | 11.9 |
| Ethylene glycol (EG) | $HOCH_2CH_2OH$ | 41.4 | 16.06 | 197 | 48.02 | 32.9 | 17.0 | 11.0 | 26.0 |
| N-Methyl-2-pyrrolidone (NMP) | $C_5H_9NO$ | 32.55 | 1.65 [148] | 203 | 40.21 | 23.0 | 18.0 | 12.3 | 7.2 |
| Benzyl alcohol | $C_6H_5\text{-}CH_2OH$ | 11.92 (30 °C) | 5.47 | 205 | 36.8 [149] | 23.8 | 18.4 | 6.3 | 13.7 |
| Formamide | $HCONH_2$ | 111 | 3.34 | 210 | 57.03 | 16.6 | 17.2 | 26.2 | 19.0 |
| 1,3-propanediol (PDO) | $HO(CH_2)_3OH$ | 35.1 | 42.3 [126] | 214 | 53.125 [150] | 16.8 | 13.5 | 23.2 | 72.5 |
| Diethylene glycol | $(HOCH_2CH_2)_2O$ | 31.82 | 30.2 | 245 | 44.82 | 29.1 | 16.6 | 12.0 | 20.7 |
| Glycerol | $HOCH_2\text{-}CHOH\text{-}CH_2OH$ | 46.53 | 934 | 290 | 64 (20 °C) [151] | 36.1 | 17.4 | 12.1 | 29.3 |
| Nafion N115 | N/A | N/A | N/A | N/A | N/A | 23.5 [152] | 17.4 | 12.5 | 9.6 |
| Nafion N117 | N/A | N/A | N/A | N/A | N/A | 19.9 [153] | 15.1 | 8.9 | 9.4 |
| Aquivion E87-125 | N/A | N/A | N/A | N/A | N/A | 23 [130] | 17.0 | 10.1 | 11.8 |

* Hildebrand solubility parameter ($\delta$) = Hansen solubility parameter ($\delta_{total}$). ** D: dispersive forces, P: polar forces, H: forces caused by H-bonds.

## 5. Catalyst Ink: Effect of Its Preparation on Its Microstructure and Macroscopic Properties

Except for the ink formulation, its preparation process also exerts an impact on its microstructure and macroscopic properties, including the introduction order of ingredients and the dispersion process.

### 5.1. Order of Ingredients Mixing (Adding Order)

Although only a few studies on the effect of the order of ingredients mixing during the ink preparation process exist, it has an influence on the size of agglomerates in the catalyst ink and the material stability through affecting the interaction sequence of catalyst, ionomer, and solvent [154,155]. It has been reported that directly mixing Nafion dispersion solution (the commercial dispersion solution contains the water/1-propanol mixed solvent) and Pt/C likely detaches Pt nanoparticles from carbon due to the triggered combustion reaction that would degrade carbon support [155]. To avoid such degradation, Pt/C is usually mixed with sufficient water prior to introducing Nafion dispersion solution [154]. Furthermore, Carine et al. found the sequence of ingredients mixing has an impact on the ink viscosity and the size of agglomerates. If keeping other conditions consistent, pre-mixing Pt/C, water, and ethanol and then adding ionomer dispersion solution will decrease the size of agglomerates and achieve a more viscous ink and hence a more homogeneous CL [154]. Therefore, to optimize microstructure and macroscopic properties of the ink prepared, attention needs to be paid to the order of ingredient mixing as well.

### 5.2. Dispersion Process

When adding a catalyst and ionomer into a solvent, the catalyst and ionomer tend to remain as large agglomerates, and the interaction between them is weak. Therefore, the ink dispersion process is necessary and determines the size and size distribution of agglomerates and interaction between catalyst and ionomer, as well as the reproducibility, viscosity, thixotropy, or stability of the ink [29,156–158]. The main purpose for the ink dispersion process is to break down the initial large agglomerates, homogeneously disperse catalyst and ionomer in the solvent, and boost the interaction between catalyst and ionomer, which are essential for enhancing the Pt utilization. To date, the typical dispersion methods include ball milling, mechanical stirring, and ultrasonication (tip sonication or bath sonication). Ultrasonication and ball milling are the most widely used ink dispersion methods. For the ink with low viscosity or low solid concentration, tip sonication, bath sonication, or their combination, and ball milling are applicable; however, for a viscous ink or an ink with high solid concentration, ball milling may be more suitable, e.g., ca. 10 wt.% solid concentration [24,28]. Furthermore, the hydrodynamic cavitation method [24], the planetary mixer [154], or high-pressure homogenization [158] have been applied to disperse catalyst ink recently. Ball milling, mechanical stirring, and ultrasonication as well as their pros and cons have been well summarized in a recent review [29]. This review only pays attention to the effects of the dispersion process on the microstructure and macroscopic properties of the ink and hence the resulting CLs.

(1)　The effect on the ink microstructure: Appropriate dispersion process could effectively break up the large agglomerates, reduce the size of agglomerates, and improve the dispersion of catalyst and ionomer in solvent, which will maximize TPB, avoid the formation of cracks, as well as improve the pore volume of CL and thus the mass transport [24,156] (Figure 9b,e). However, if the dispersion process is insufficient, the large agglomerates in the catalyst ink cannot be broken down, leading to an inhomogeneous CL or even causing cracking in the CL (Figure 9a,d) [24,154]. The large agglomerates in the CL would increase the proton, gas, and water transport resistance. Ionomer molecules mainly adsorb on the exterior surface of the catalyst agglomerates and barely penetrate the catalyst agglomerates, especially the large catalyst agglomerates [156]. Therefore, the Pt inside the large agglomerates cannot access ionomer, increasing the proton transport resistance [156]. Due to the low exterior-

surface-to-volume ratio, a thick ionomer film may be formed outside the large catalyst agglomerates, increasing the gas transport resistance through the ionomer film [156]. Further, the long gas transport distance in the interior of the large agglomerates also has an adverse effect on the gas and water transport [156].

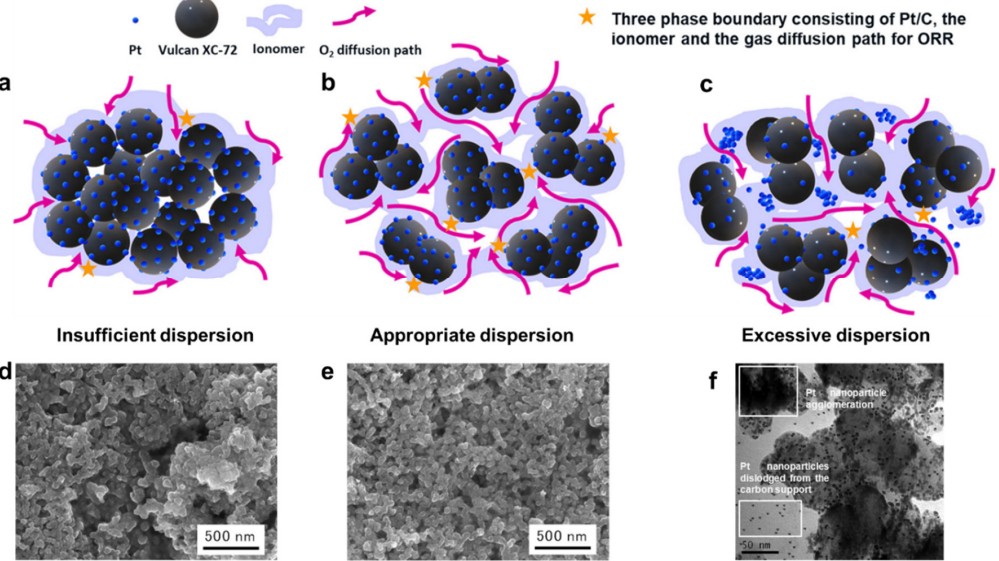

**Figure 9.** The structures of Pt/C-Nafion ionomer and CL with insufficient (**a**,**d**), appropriate (**b**,**e**), and excessive (**c**,**f**) dispersion. ((**a**–**c**) Modified with permission from [156], Copyright (2019), American Chemical Society; (**d**–**e**) Reprinted with permission from [24], Copyright (2019), American Chemical Society; (**f**) Reprinted from [159], Copyright (2014), with permission from Elsevier).

(2) The effect on the reproducibility, viscosity, or thixotropy of ink: The insufficient dispersion process decreases the reproducibility of the ink and hence the CLs made of the ink [159]. As mentioned in Section 3.2, the ink viscosity relies on its microstructure, e.g., the size of agglomerates. Therefore, the dispersion process influences the ink viscosity. For example, the large agglomerates in the catalyst ink are dispersed as the ball milling time or the sonication amplitude increase, increasing the ink viscosity [28]. When the agglomerates are broken down to a certain size or reach a certain hardness, they are not dispersed further as the ball milling time or the sonication amplitude increase; thus, the ink viscosity no longer changes. The viscosity of the ink prepared by ball milling exceeds the case of ultrasonication due to the better dispersion degree [28]. It was also found that when decreasing the shear rate to zero, the viscosity of the ink prepared by ball milling exhibits a faster reaction, i.e., an excellent thixotropy, than the case of ultrasonication [28].

(3) The effect on the stability of ink: Appropriate and effective dispersion process could form a homogeneous catalyst ink, leading to higher colloidal stability [158,160]. However, a too aggressive dispersion process, e.g., long-time ultrasonic irradiation or ball milling, or high ultrasonic power, will cause the degradation of catalyst and ionomer, e.g., the detachment, aggregation, or Ostwald ripening of Pt nanoparticles (Figure 9c,f), as well as the degradation or decomposition of ionomer [159,161].

## 6. CL Formation

In addition to the composition and preparation process of the catalyst ink, the CL formation process, namely, depositing catalyst ink onto a substrate by ink coating techniques and then drying to form a CL, also regulates the morphology and structure of CL, e.g., thickness [162], uniformity [163], porosity [38,162], cracking [104], surface roughness [38], ionomer distribution [41]; thereby impacting the performance and durability of PEM fuel cell. To fabricate a well-desired CL, the macroscopic properties of the pre-

pared ink, the structure and physical properties of the substrates, deposition methods and deposition parameters, as well as drying methods and drying parameters should match.

### 6.1. Substrate

The substrates onto which the ink is deposited include proton exchange membrane, GDL, and decal substrate. They have different surface structures, e.g., roughness [75], and properties, e.g., contact angle [164], friction coefficient [164], or wetting and permeation of solvent, which influence not only the requirements on catalyst ink, selection of deposition technique, or drying conditions, but also the interface between the CL and membrane or the CL and GDL, as well as the production scale. There are three pathways that can be used to fabricate MEAs based on the substrates used (Figure 10 and Table 3) and are summarized as follows.

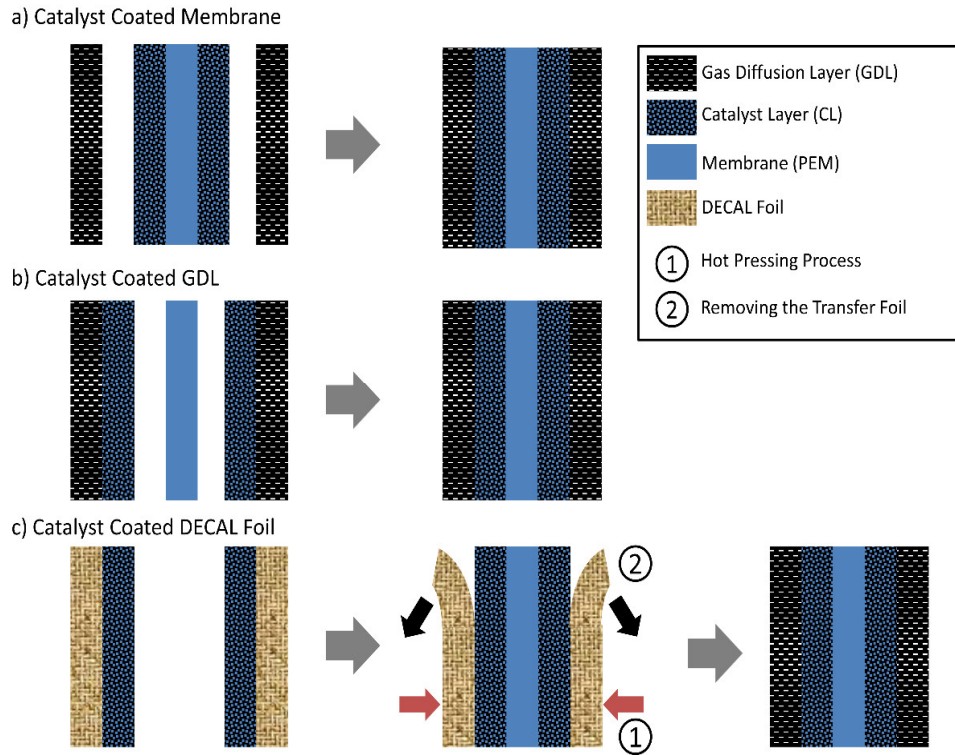

**Figure 10.** Three ways of fabricating catalyst coated membranes.

**Table 3.** Comparison of CCM, GDE, and DTM.

| Methods | Advantages | Disadvantages |
|---|---|---|
| CCM | • Good performance and durability<br>• High catalyst utilization<br>• Improved interfacial contact between CL and membrane | • Membrane swelling or wrinkling due to the direct contact with ink during manufacturing process<br>• Challenging for roll-to-roll manufacturing process<br>• (These challenges may be solved in the future with the improvement of the membrane) |
| GDE | • Simplicity<br>• Suitable for roll-to-roll manufacturing process | • Limited interfacial contact between CL and membrane<br>• Limited catalyst activation<br>• Fabricating the MEAs with low Pt loading is still challenging |
| DTM | • Overcome the membrane swelling or wrinkling<br>• Logically suitable for roll-to-roll manufacturing process | • The production cost<br>• Limited interfacial contact between CL and membrane |

(1) Catalyst coated membrane (CCM) method: Catalyst ink is deposited onto both sides of the membrane, directly forming CLs on the membrane, known as CCM. The MEA is achieved through hot-pressing the CCM sandwiched between two GDLs (Figure 10a). (Industrially, GDLs may be glued to the edges of a frame and then CCM is sandwiched between two GDLs when assembling the fuel cell stack. Contact of CCM with GDLs is realized by stack compression.) The MEA produced by the CCM method commonly has an excellent interfacial contact between the CL and membrane and a higher Pt utilization due to directly applying the CL onto the membrane, resulting in higher power and durability. Membrane swelling or wrinkling occurs during the coating process, due to direct contact with the solvent in the ink. This can be mitigated to some extent a small or moderate scale fabrication (e.g., in the lab-scale fabrication or in the moderate production where many companies use CCM method at present) by mechanically fixing the membrane (e.g., on a vacuum platform or in a die) or shorten the interaction time between the membrane and solvent via heating and selecting spray coating to deposit the ink. However, although the membrane swelling and the solvent penetration could be decreased through selecting the appropriate solvent [165], it is still a challenge to directly deposit the ink onto the membrane in a continuous, scale production, e.g., roll-to-roll manufacturing process. This is attributed to the difficulty to mechanically stabilize the membrane during the coating process unlike the case of lab-scale fabrication. Moreover, the catalyst ink is applied onto the membrane at once during the roll-to-roll coating, which means a long interaction time between the membrane and solvent before complete evaporation. With the improvement of the membranes, the challenges CCM method faces in the roll-to-roll manufacturing process may be solved in the future.

(2) Gas diffusion electrode (GDE) method: Catalyst ink is deposited onto the MPL, directly forming the CL on a GDL, obtaining a GDE. The MEA is achieved through hot-pressing the membrane sandwiched between two GDEs (Figure 10b). The membrane swelling or wrinkling could be avoided for the GDE method due to indirect contact with the solvent. However, the GDE method typically offers a limited interfacial contact between the CL and membrane compared to the CCM method, likely resulting from the difference in the surface roughness of MPL and membrane. The MPL has a higher origin surface roughness than the membrane (Figure 11(a1,b1)) [75]. When applying the CL onto the membrane, the CL surface facing the membrane is similar to the membrane surface due to the nearly conformal deposition, forming an excellent contact between the CL and membrane. When applying CL onto MPL, the CL surface facing the membrane is uneven and has an increased roughness compared to the MPL (Figure 11(b2)), making it hard for the membrane to contact the entire CL surface and thus causing gaps (similar to the case shown in Figure 11c,d) [166]. This situation will be more serious when the MPL's surface roughness increases [166]. To improve the interface between the CL and membrane and reduce the interfacial resistance, coating an extra ionomer layer on the top of the GDE was often applied [166,167]. It has been found that the critical ionomer loading increases as the surface roughness of MPLs increases [166]. Further, the hot press is also necessary for fabricating the GDE-based MEAs, which is not necessary for the CCM-based MEAs [167]. Furthermore, catalyst particles may penetrate the pores or cracks of MPL, reducing the catalyst activation and thereby the performance of MEA. The penetration degree depends on the deposition method due to the difference in the viscosity of the used ink or the drying rates for various coating techniques [38].

(3) Decal transfer method (DTM): CL is deposited on an inert decal substrate first and then transferred to the membrane through the hot press, achieving the CCM, known as DTM (Figure 10c). This method was developed to overcome the drawbacks of the CCM method by avoiding the direct contact of the membrane with the catalyst ink. Although the surface of the decal substrate is similar to the membrane, the CL surface facing the membrane is still uneven but has a lower roughness than that

produced by the GDE method [75]. From an economy perspective, direct coating (CCM and GDE) is advantageous; however, decal transfer is not desirable due to the increased production cost caused by the transfer process, the usage of decal substrate, and the waste of catalyst and ionomer due to the incomplete transfer caused by the inappropriate process parameters. To increase the transfer yield and decrease the hot-press temperature simultaneously, considerable efforts have been devoted, e.g., using swelling agents to pre-treat the CL before the hot press [168], pre-depositing a breaking layer composed of carbon, or a carbon and Nafion mixture onto the decal substrate [169], selecting appropriate decal substrate [164], or depositing an extra ionomer layer on the top of CL before the hot press [164]. Nevertheless, it is difficult to completely remove the swelling agents since they usually have a high boiling point, e.g., 1,5-pentanediol [168]; the break layer will increase the interfacial resistance between the CL and GDL. Samaneh et al. [164] developed an effective low-temperature decal transfer method with complete transfer yield and the improved interfacial contact between the CL and membrane, through investigating the type of decal substrates, hot-pressing condition, as well as an extra Nafion layer on the top of CL (faces the membrane) and the Nafion ionomer loading. It was found that fluorinated ethylene propylene film is a good decal substrate and superior to the commonly used polytetrafluoroethylene film due to its low friction coefficient and low contact angle. If depositing an outer Nafion layer with a Nafion loading of 0.2 mg/cm$^2$ on the CL before the hot press, the CL can be completely transferred to the membrane at a low hot-pressing temperature of 130 °C, and the extra Nafion layer could improve the interfacial contact between the CL and membrane, which is similar to the case of GDE.

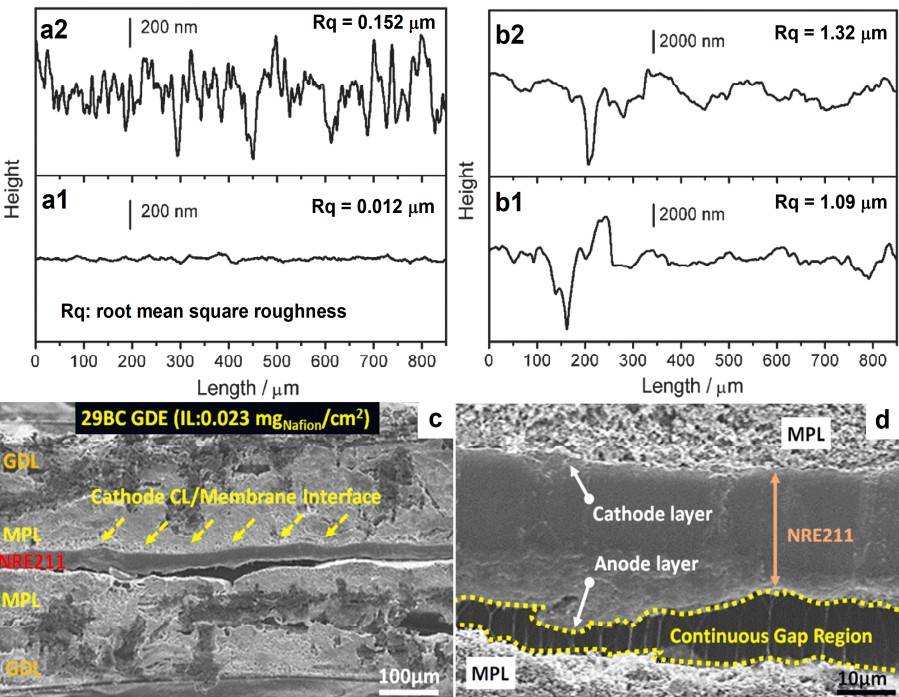

**Figure 11.** Stylus profilometry traces of Nafion membrane (**a1**), CL deposited on membrane (the surface facing the GDL) (**a2**), GDL (the surface of MPL facing CL) (**b1**), and CL of GDE (the surface facing the membrane) (**b2**) (Reprinted from [75] with permission of John Wiley & Sons); (**c**,**d**) SEM images of cross-section view of the MEA. Cathode: a good CL/membrane interface due to the ionomer overlayer deposited on GDE; Anode: the observed gaps at the interface of MPL and CL which was deposited on the membrane, due to the high MPL surface roughness (root mean square surface roughness: 15.85 μm) (Reprinted with permission from [166]. Copyright (2019) American Chemical Society).

The CCM-based MEAs, GDE-based MEAs, and DTM-based MEAs generally have different performances due to the different substrates used [167]. The differences between them are more pronounced when the catalyst loading is low because the CLs are more sensitive to the underlying substrate structure with a decrease in the CL thickness or the catalyst loading. In particular, the differences between the CCM-based MEAs and GDE-based MEAs appear extremely pronounced due to the different surface roughness and the catalyst loss caused by the penetration into the porous MPL [167]. However, there is not a consistent conclusion where MEA exhibits the best performance in the literature, because the MEA performance is also sensitive to the coating techniques, catalyst loading, and hot-pressing conditions. The CCM-based MEAs exhibit a superior performance in some studies due to excellent interfacial contact between the CL and membrane [167], while the GDE-based MEAs perform better in others [75].

*6.2. Coating Process*

The ink coating techniques commonly used include spray coating, brush coating, doctor blades coating, screen printing, inkjet-print coating, slot-die coating, or gravure coating [29,38,170]. Spray coating is the coating technique commonly used in small or moderate scale and usually used in the lab for evaluating the performance of the advanced materials. At the end of 2014, Toyota released its first generation of PEM fuel cell vehicles to the market; at the end of 2020, it released its second generation of PEM fuel cell vehicles with improved performance to the market, including the increased driving range (has increased the world record distance to 1360 km with one hydrogen fill recently) or increased stack volume power density (5.4 kW/L). In recent years, the other vehicle companies, e.g., Honda and Hyundai, have released their version of PEM fuel cell vehicles to the market. Recently, the application of PEM fuel cells in heavy-duty vehicles has also attracted significant attention [171,172]. As the demand for the application of PEM fuel cells increases and is expected to increase substantially in the coming years, the cost of MEAs production needs to be reduced further and the MEAs production yield needs to be increased. These need to reduce the process steps of MEAs and the reliance on manual labor (the small/moderate-scale coating techniques often need highly trained labor and are labor-intensive), as well as realize the continuous production. Therefore, roll-to-roll coating method attracted attention and has been applied to deposit CLs recently. The working principles, related to coating parameters, or pros and cons of various coating techniques have been well summarized in recent reviews [29,170]. Therefore, in this review, the attention is only paid to the spray coating and the roll-to-roll manufacturing process coupled with slot-die coating or gravure coating. A brief description on their development and the effect of the relevant coating parameters on the morphology or structure of CLs is presented as follows.

(1) Spray coating: Spray coating works mainly through applying high energy (e.g., high-pressure gas turbulence or ultrasonic vibration) to break the large catalyst ink droplets into tiny droplets finally deposited onto the substrate. It has several merits, such as relatively high uniformity of the CLs and relatively precise control on the catalyst loading since the CL is deposited layer by layer. However, the spray coating is a slow process, only suitable for depositing CLs on a small or moderate scale [165]. Therefore, spray coating is more suitable for fabricating the CLs in the lab used to evaluate the performance of the advanced materials due to the relatively excellent control on the CL uniformity and the catalyst loading. The most used spray coating method is the gas-assisted spray. To form a fine ink mist with well-defined droplets size, ultrasonic spray and electrospray were developed [104,162,163]. Compared to the CL deposited by the gas-assisted spray, the electrospray-coated CL is much more porous and has an increased ECSA value due to the fine ink droplets formed by the coulomb repulsion [162].

During spray coating, some spraying parameters influence the morphology and structure of the CL and hence the performance of the cell, such as the nozzle height [104], electrospray mode [67], or spraying voltage [163]. While the thickness and porosity are

independent of the ultrasonic nozzle height, the CL surface may crack when the distance between the nozzle and substrate is too short (e.g., 3.5 cm) due to the presence of excess liquid ink on the substrate [104]. When the injection rate is fixed and the intensity of the electric field (determined by the working distance and the spraying voltage) is within a certain range, a stable cone jet mode can be formed, where the large ink droplets can be effectively broken into fine droplets due to the sufficiently strong coulomb repulsive force [163]. In the cone mode, the coulomb repulsive force will increase as the electric field intensity increases, which is beneficial for creating the fine droplets; therefore, the agglomerate size in the CL is reduced and a CL with more uniform pore distribution would be achieved [163]. Seonghun et al. [67] found that the ionization mode of electrospray has an influence on the morphology of ionomer and then impacts the ionomer film covering catalysts. The CL fabricated by the electrospray with negative ionization mode shows a decreased thickness of ionomer film covering catalysts, which will reduce the local gas transport resistance through the ionomer film.

(2) Roll-to-roll manufacturing process: Roll-to-roll manufacturing process is a continuous and scalable production method (Figure 12a). The CL fabrication speed would be increased by over 500 times more than the lab-scale spray coating when using the roll-to-roll coating method [165].

For the roll-to-roll method, catalyst ink is deposited on the substrate at once. Therefore, the ink needs to have a relatively high solid content and relatively high viscosity, different from spray coating, which commonly needs an ink with low solid content and low viscosity. GDE method and DTM are logically more suitable for roll-to-roll production than the CCM method as the GDL and decal substrates are more rigid than the membrane. Compared to decal film, the GDL is more appropriately used as the substrate for roll-to-roll manufacturing process from economic perspective [38,39,41]. As mentioned previously, it is often necessary to apply an extra ionomer overlayer on the surface of the CL to improve the interface contact of the CL and membrane when choosing the GDL as substrate. However, this will increase the process steps, decreasing the advantage of the GDE method for mass production. To avoid depositing an extra ionomer layer and improve the interface contact between the CL and membrane, Mauger et al. [41] studied the effect of ink coating techniques and drying temperature on the distribution of ionomer through the CL in order to directly create a CL with an ionomer-rich surface during the fabrication process. They observed that increasing the solvent evaporation rate could enrich the ionomer at the top of the CL (facing the membrane) during the roll-to-roll manufacturing process coupled with slot-die coating, improving the performance of the MEA [41].

Slot-die coating and gravure coating are widely used ink coating techniques of the roll-to-roll method (Figure 12b,c). For slot-die coating, the catalyst loading and the homogeneity of the coated ink film could be controlled by adjusting the solid content in ink, the gap height, or the feed ink volume [39]. For gravure coating, the thickness and quality of the CL deposited are affected by the cylinder rotation speed and engraved volume [38]. The engraved volume and the cylinder rotation speed provide a coarse and relatively fine control on the catalyst loading, respectively; the catalyst loading increases as the engraved volume or cylinder rotation speed increase [38]. Increasing the rotation speed or engraved volume also enhance the coating quality and avoid the formation of visible defects, e.g., pinholes or bubbles [38]. Furthermore, it was found that the penetration degree of catalyst into the pores or cracks of MPL could be reduced if using gravure coating due to the relatively high viscosity of the ink used [38]. The slot-die-coated or gravure-coated CLs show significantly larger porosity or thickness than the spray-coated CLs, because of the concentrated ink and the application of the ink at one time [38]. Compared to the gravure-coated CL, it was found that the slot-die-coated CL may have a higher Pt utilization and a better performance at various RH, especially at low RH [173]. These may be attributed to the increased time at high shear, leading to the homogeneous distribution of catalyst and ionomer in the slot-die-deposited ink film, and hence the improved ionomer-catalyst contact and lower tortuosity or larger conduction pathways in the CL [173].

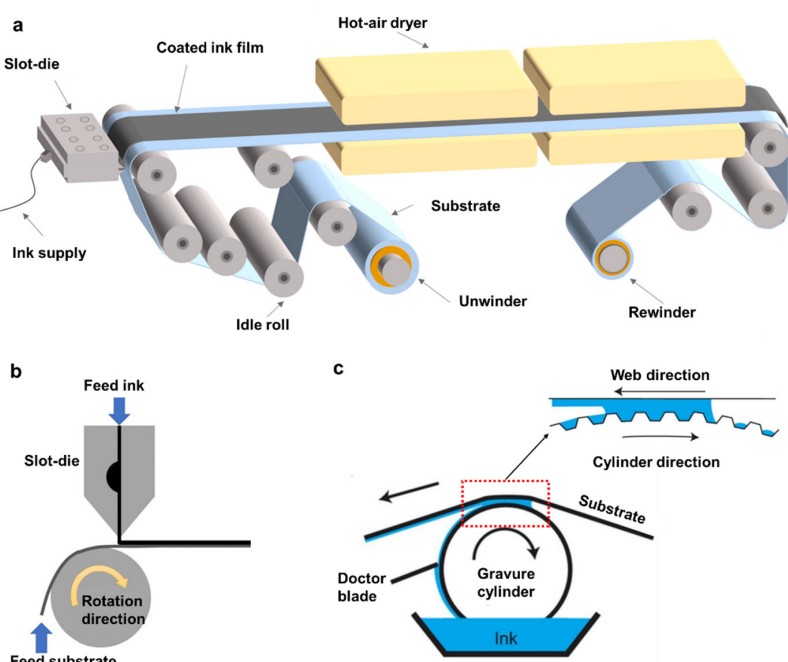

**Figure 12.** (**a**) Schematic of the roll-to-roll manufacturing process coupled with slot-die coating, (Reprinted from [165], Copyright (2020), with permission from Elsevier), (**b**) illustration of slot-die coating (Adapted from [39], Copyright (2019), with permission from Elsevier), (**c**) illustration of gravure coating (Adapted from [38]).

The application of the roll-to-roll manufacturing method in the CLs fabrication is still in the early stage. Therefore, to apply it in the practical production of CLs (especially, the CLs with low Pt loading or the thin CLs) as soon as possible, more efforts are needed to optimize the formulation and preparation of the catalyst ink, coating parameters, or the drying conditions, and study their effect on the coating quality as well as the morphology and structure of the CL.

*6.3. Drying Process*

If the formulation and preparation processes of the catalyst ink, as well as the coating process are fixed, the drying process of the ink film deposited on the substrate will determine the morphology and structure of the CL. There are several influencing factors that should be considered, e.g., the drying techniques, evaporation conditions, solvent physical properties, or solid concentration in the ink. The drying methods that have been used to dry the CLs previously include freeze drying, vacuum drying, heated substrate drying, oven drying, or forced-air oven drying [41,174]. It was reported that compared to the vacuum-dried CL or oven-dried CL, the freeze-dried CL shows the improved porosity and ionomer distribution, enhancing the gas transport [174]. However, the freeze drying technique is not wildly used to dry the CL since the drying process is complicated, costly, and dangerous due to the usage of liquid nitrogen. There are primarily two major factors influencing the evaporation rate, i.e., the physical properties of the solvent used (the boiling point and vapor pressure, as discussed previously) and the evaporation condition (temperature, ambient pressure, or RH). The solid concentration of catalyst ink also has an influence on the drying dynamic and thus leads to the different microstructures of the dried CLs. When dripping a droplet of ink (1 μL) onto the substrate, the ink with a relatively high solid concentration (e.g., 1.59 wt.%) has a smaller contact area (spherical crown, diameter 6 mm, height 70.3 μm) on the substrate, resulting in a smaller surface exposed to air and hence a lower solvent evaporation rate (total evaporation time 28 s), than that with a low solid concentration (e.g., 0.2 wt.%, spherical crown, diameter 8.3 mm, height 37 μm, total evaporation time 18 s). After the solvent is completely evaporated, the resulting CL surface

becomes rougher and the distribution of catalyst and ionomer in the resulting CLs becomes more inhomogeneous with decreasing the solid concentration [40].

Although the structure evolution of CLs during the drying process is still unclear, there are some processes possibly dictating the distribution of materials or governing the formation of the CL structure, e.g., solvent evaporation, internal particles flow (or migration) in the ink film due to concentration and temperature gradients induced by solvent evaporation, sedimentation resulting from gravitational force, agglomeration, assembly, packing, or shrinking [41,175,176]. To understand the drying process, Suzuki et al. monitored the drying process through combining in-situ visualization using atmospheric SEM with weight test and studied the effect of drying rate (drying temperature or humidity) on the porosity and pore size of the CLs [56]. It was found that there are two packing steps occurring in the ink film with solvent evaporation, i.e., sedimentation, and then shrinking. In the initial period, the solid concentration in the ink film increases and the ink film thickness decreases rapidly over evaporation [56] (Figure 13). When the solid concentration reaches a critical value, the particles in the ink film contact each other and form a network, i.e., a solvent-saturated CL, and then the drying process enters the second stage. The porous structure is formed gradually, and the thickness of CL slightly decreases over solvent evaporation in the second period, which needs a longer time than the first stage [56] (Figure 13). During this stage, the capillary force works between the particles near the drying front that moves from the surface to the inside during evaporation. The capillary force compresses the particles, and the pores shrink until the porous structure is strong enough [56]; however, the excessive capillary force may cause cracking, which will be discussed in the last part of this section. A rapid evaporation rate will shorten the first and second drying process, leading to an increased porosity, but having little impact on the pore size [56]. The increased porosity of CL with an increase in evaporation rate, namely, an evaporation-dominated process, was observed in Mauger's studies as well [41]. However, Mauger et al. found that the drying rate has an obvious influence on the secondary pore size, which decreases with decrease in the solvent evaporation rate, since the particles in the ink film have enough time to migrate and pack densely [41]. The inconsistent results are likely due to the different drying temperatures, 20–80 °C in Mauger's studies vs. 20–40 °C in Suzuki's studies. Moreover, it has been found that the evaporation-dominated consolidation leads to the CL with the smaller-particles-rich surface, i.e., the ionomer-rich surface due to the smaller size of the non-adsorbed ionomer agglomerates than catalyst-related agglomerates [41]. During the shrinking process, Kusano et al. found that the size of catalyst agglomerates and the thickness of Nafion shell around the catalyst agglomerates will decrease over evaporation, due to the exclusion of the solvent molecules from primary pore and the solvated Nafion ionomer shell [175].

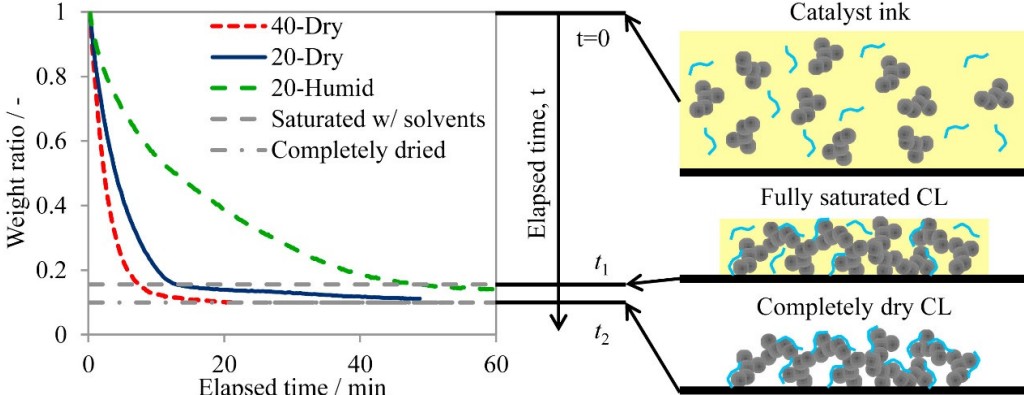

**Figure 13.** Weight change of the ink film at different drying conditions (20 or 40 °C, dry—0% RH, humid—80% RH) and the schematics of the structure change from the ink film to the dried CL (Reprinted from [56], Copyright (2016), with permission from Elsevier).

Cracking is a common issue during the CL formation, especially when applying a thick ink film on the substrate at one time, e.g., screen printing [66] or slot-die coating [79]. Although the CLs with guided cracks generated by starching the prism-patterned CCM exhibited enhanced water transport at the cathode and thus increased performance at the mass polarization region [177,178], it was reported that the unguided cracks formed during the CL formation process have an adverse impact on the durability, such as causing flooding at the cracks [179,180]. Therefore, the cracks should be avoided in the CLs as much as possible during the drying process. Based on the previous research, the formation of cracks is primarily because of the excessively strong drying stress caused by capillary force or the swelling and then shrinking of the membrane. When the solvent molecules at the upper surface evaporate, the particles near the surface are no longer covered by a plane solvent film. Menisci develops between the particles along with the build-up of capillary force, causing capillary shrinkage [181]. If the capillary shrinkage is severe or imbalanced, the cracks will be formed to release the growing tensile stresses. Although the nucleation of cracks is not yet fully understood, there may be several factors promoting cracking.

(i)　Extremely quick evaporation of the solvent: Too fast solvent vaporization rate, e.g., the solvent with low boiling point, easily causes much difference in the drying speed between the surface and interior, resulting in the increased drying stress and then cracking in the dried CL [42,66].

(ii)　Weak interaction between the ionomer and catalyst [125,175]: Good affinity between ionomer and catalyst enhances the adhesion strength between particles because ionomer acts as a binder, which could avoid cracking. Furthermore, the deformation of ionomer would dissipate the drying stresses in some degree due to the low Young's modulus of ionomer, decreasing the risk of cracking [74].

(iii)　Defects: The defects in the catalyst ink film, e.g., small voids, bubbles, pinholes, uneven film, large catalyst–ionomer agglomerates, or self-organized free ionomer, could induce stress concentration, increasing the risk of cracking [24,81,125]. For example, the large agglomerates in the catalyst ink will induce the non-uniform surface tension during the CL formation, leading to cracking in the dried CL. The large agglomerates may result from the poor dispersion of catalyst ink [24], or the hydrophobic compounds formed from the alcohol oxidation catalyzed by Pt, e.g., 1,1-dipropoxypropane and propyl propionate, which promote the aggregation of catalyst agglomerates in the ink [81]. Therefore, the well-dispersed ink with small and homogeneous agglomerates and depositing an even and uniform ink film on the substrate will be helpful to suppress cracking.

## 7. Future Direction and Prospects

To supply a required performance, rapid electron, proton, gas, and water transport are necessary in the networks of the CL composed of Pt/C, ionomer, and pore. This is closely related to the CL multiscale structure, which is formed during the fabrication process and thus determined by the composition and preparation process of the catalyst ink and the CL formation process. After continuous research and development, the effect of the formulation and preparation processes of catalyst ink on its microstructure and macroscopic properties and then on the CL structure as well as the effect of the substate, coating process, and drying process on the formation, morphology, and structure of the CL has been understood to some degree. However, for the time being, the relationship between the formulation and preparation processes of catalyst ink or the CL formation process and the multiscale structure of CL and further the performance and durability of PEM fuel cell is still unclear. Much more effort should be devoted to thoroughly clear this relationship through investigating the effect of the composition and preparation process of the catalyst ink, the substrate, coating process, and drying process on the morphology and structure of the CL.

(1)　The catalyst ink microstructure: To uncover how the composition and preparation process of catalyst ink affect the structure of the CL made of the ink, it is necessary

to accurately characterize the ink microstructure. The characterization techniques used in the literature have been summarized in Section 3.1. However, because of the opaque, dynamic, and complicated properties, it is difficult to directly characterize the ink microstructure. To date, the characterization techniques used to measure the ink microstructure commonly need to modify its initial concentration, e.g., dilution (e.g., DLS or ELS) or consolidation (e.g., TEM and SEM). Furthermore, some techniques only measure the local microstructure, e.g., TEM or SEM. The results cannot truly assess the microstructure of catalyst ink, a complex system, or are even misleading at times. Hence, it is necessary to introduce or develop some advanced characterization techniques to accurately evaluate the microstructure of catalyst ink without any modification, e.g., XCT, SAS, or CV-SANS. Furthermore, since the macroscopic properties of a catalyst ink are usually determined by its microstructure, the catalyst ink microstructure could be coarsely inferred through testing its macroscopic properties, e.g., the viscosity or rheologic behavior. The microstructure of catalyst ink is primarily governed by the interaction between catalyst, ionomer, and solvent, which could be studied by model and numerical simulations, e.g., molecular dynamic simulation [118] or discrete element method [40]. Therefore, combining the modeling and physical characterization of the ink might allow us understanding the ink microstructure, to optimize the ink formulation and dispersion process.

(2) Inconsistent conclusions reported in the literature: Because the microstructure and macroscopic properties of catalyst ink are determined not only by the types of catalyst, ionomer, and solvent(s) as well as the ratio between them, but also by the preparation process. If only varying one of the variables, such as merely changing catalyst or using different dispersion methods, the prepared catalyst inks may show different microstructure or macroscopic properties. The morphology and structure of the CL made from catalyst ink are co-governed by the microstructure or macroscopic properties of the catalyst ink, the substrate, coating process, and drying process. This makes the influence of the composition and preparation of catalyst ink on the multiscale structure of the CL extremely complicated and increases the level of difficulty uncovering their relationship. Finally, the CL structure, hot-pressing process, and the fuel cell assembly will co-determine the performance and durability of PEM fuel cell [167,170]. For example, if using dipropylene glycol (DPG)/1-propyl alcohol (NPA) and water mixtures as the solvent, the ionomer agglomerates with moderate size close to that of Pt/C agglomerates in the catalyst ink not only form a connected ionomer network, but also maintain the adequate porosity in CL [23,25], offering an improved PEM fuel cells performance. However, in another, the PEM fuel cells with a high performance need large Nafion agglomerates in the IPA/water-based catalyst ink, which leads to a thinner ionomer film on the catalyst surface [26]. In the IPA-, dimethyl sulfoxide (DMSO)-, N-Methyl-2-pyrrolidone (NMP)-, ethylene glycol (EG)-, or propylene glycol (PG)-based catalyst ink, the solvent that could make ionomer uniformly distribute will allow for good PEM fuel cells performance, due to the homogeneous coverage on catalyst agglomerates [126,127]. The reasons why confused conclusions were often reported in the literature are because there are many factors simultaneously influencing the relationship between the formulation and preparation of catalyst ink or the CL formation process and the CL structure and further the performance and durability of the PEM fuel cell. If the other conditions or processes are inconsistent except the main variable in different studies, different or even opposite conclusions may be obtained.

(3) Stability of catalyst ink during storing and reversibility of the aged ink: It is impossible in most situations that the prepared catalyst ink is applied immediately or completely consumed in a short time; in other words, the prepared ink needs to be stored before appliance. To guarantee the consistency of the structure and performance of the CLs fabricated at different periods, the composition and microstructure of catalyst ink should be stable. However, it is generally hard to avoid the material degradation, agglomeration, or sedimentation in the catalyst ink during storage. Consequently,

the CLs produced in different periods likely show different structures and hence performances due to the altered microstructure and macroscopic properties of the ink over time, which will have a terrible influence on the performance and durability of PEM fuel cell stack. While the material degradation, especially that caused by Pt catalysis, is hardly inhibited and recovered, the agglomeration and sedimentation are able to be inhibited, retarded, or recovered to some degree. To keep it uniform, Hoffmann et al. permanently stirred the catalyst ink with a magnetic stirrer after dispersion [74]. Stirring may prevent sedimentation to some extent but it is unclear if the agglomeration can be inhibited since stirring cannot effectively break down the large agglomerates in the catalyst ink. Thus, it should be taken into account how to effectively and quickly redisperse the aged ink or recover its uniformity when applying it, and the extent of reversibility.

(4) Studies on the catalyst ink with high solid concentration: At present, most studies on the composition and preparation process of catalyst ink focus on the cases of low solid concentration. Nevertheless, the roll-to-roll coating method, suitable for continuous, scalable production, always requires a slurry with high solid concentration, which may possess different microstructure and macroscopic properties from the ink with low solid concentration. For example, the thin ink nearly shows Newtonian feature; therefore, its rheology may barely impact the ink deposition process and thus the formation of CL. However, the slurry with high solid concentration always exhibits a high shear-thinning degree, which will increase over the solid concentration. In this case, as the rheology of ink will have an important influence on the ink deposition process and the formation of CL, it is necessary to rationally design the ink rheology by modulating the ink recipe and dispersion process. Hence, more attention should be paid to investigate the impact of the ingredient and preparation on the microstructure and macroscopic properties of the concentrated catalyst ink to promote the mass production of MEAs.

(5) Drying process: While the drying process plays a vital role in the formation of the morphology and structure of CL, there are limited studies examining the evolution of ink film into a dried CL during solvent evaporation. Due to the opaque, dynamic, and complicated properties of the ink and the lack of in situ characterization techniques, it is hard to figure out how the particles in ink film move with solvent evaporation and assemble into the final porous structure as well as the nucleation mechanism of cracks. However, clearing the drying process will contribute to fabricating the CLs with a well-desired structure. Therefore, more research is necessary to make sense of the drying process and the reasons causing cracking, especially the drying process of the slurry with high solid concentration.

## 8. Concluding Remarks

Not only the materials used in the CL, but also its structure, play a significant role in the improvement of the performance and durability of PEM fuel cells and consequently lead to cost reduction. The CL structure determines the transport rate and transport probability of proton, electron, gas, and water to or from the active sites. The CL is a complicated system, composed of the continuous and intersecting networks of catalyst (Pt/C), ionomer, and pore, and possesses a multiscale structure, from microscale to mesoscale and then to macroscale. The CL is commonly fabricated through an ink-based process, i.e., homogeneously dispersing catalyst and ionomer in solvent(s), obtaining the catalyst ink, and then coating catalyst ink on a substrate, finally forming a CL after solvent evaporation. The structure of CL is formed during the manufacturing process and is dependent on the microstructure and macroscale properties of the catalyst ink, the substrate, coating process, and drying process.

According to the results reported in the literature, the commonly used catalyst ink is composed of the primary catalyst agglomerates surrounded by ionomer, non-absorbed primary ionomer agglomerates, and secondary agglomerates. It is these agglomerates

that aggregate, assemble, and pack during the drying process, forming the CL. Therefore, their size, size distribution, and shape, as well as the absorbed ionomer content on the catalyst agglomerates and the interface or interaction between catalyst and ionomer in catalyst ink will govern the CL structure. The macroscopic properties of catalyst ink include viscosity, rheology, thixotropy, surface tension, and stability, which will impact the selection of coating techniques, coating speed, coating quality and reproducibility, as well as the uniformity/integrity, thickness and wettability of the deposited ink film, eventually influencing the multiscale structure of CL.

The microstructure and macroscopic properties of catalyst ink are sensitive to the formulation and preparation processes. The former includes the types of catalyst, ionomer, and solvent(s) and their ratios; the latter includes the order of ingredient mixing and the dispersion process. The formulation influences the microstructure and macroscopic properties of catalyst ink mainly via affecting the underlying interactions between the individuals. The order of ingredient mixing determines the interaction sequence of the catalyst, ionomer, and solvent, which has an effect on the agglomerate size or the material stability. The dispersion process needs to effectively break up the large agglomerates into well-desired sizes and cannot degrade or decompose the materials.

In addition to the catalyst ink, the CL formation process influences the morphology and structure of CL as well, e.g., thickness, uniformity, porosity, cracking, surface roughness, or ionomer distribution. To achieve a well-defined CL, the macroscopic properties of the prepared ink, the structure and physical properties of the substrates, coating methods and coating conditions, as well as drying methods and drying conditions should be rationally controlled and match each other. Based on the selected substrate, there are three pathways to fabricating MEAs, i.e., CCM, GDE, and DTM. Each of them has pros and cons. Compared to CCM and DTM, GDE may be more suitable for application in mass production. Among a variety of coating techniques, spray coating is more suitable for preparing the CL used to assess the performance of the advanced materials in the lab because of the relatively high uniformity of spray-coated CLs and the relatively precise control of catalyst loading. Compared to spray coating, the roll-to roll manufacturing process coupled with slot-die coating or gravure coating shows the advantage in the continuous, scalable production of MEAs. During the drying process, some behaviors, e.g., solvent evaporation, internal particle migration in the ink film, sedimentation, aggregation, assembly, packing, or shrinking, will determine the distribution of materials and impact the morphology or structure of the dried CL, which is sensitive to the drying methods, solvent evaporation rate, and the solid concentration. The formation of cracks likely results from the high solvent evaporation rate, the poor interaction between the catalyst and ionomer, or the defects in the ink film.

Finally, to clear the relationship between the formulation and preparation processes of catalyst ink or the CL formation process and the multiscale structure of CL and further the performance and durability of the cell, perspectives on the challenges and future directions are outlined. More attention should be paid to the characterization of the catalyst ink microstructure without modification, stability of catalyst ink during storage and reversibility of the aged ink, studies on the catalyst ink with high solid concentration, as well as the drying process. Furthermore, when investigating the effects of a variable on the CL structure, the other conditions or processes should be consistent to avoid achieving inconsistent conclusions.

**Author Contributions:** Writing—original draft preparation, H.L.; writing—review and editing, H.L., L.N., N.Z. and X.L.; supervision, X.L.; project administration, X.L.; funding acquisition, X.L. All authors have read and agreed to the published version of the manuscript.

**Funding:** H.L. and X.L. acknowledge the financial support from the Canadian Urban Transit Research and Innovation Consortium (CUTRIC) via project number 160028 and the Natural Sciences and Engineering Research Council of Canada (NSERC) via a Discovery Grant. L.N. and N.Z. acknowledge the financial support of the Ministry of the Environment, Climate, Protection and the Energy Sector Baden-Württemberg within the project HyFab-Baden-Württemberg, contract no. L75 20113; and The

German National Innovation Program Hydrogen and Fuel Cell Technology and The Federal Ministry of Transport and Digital Infrastructure within the project "OREO" grant number 03B11018A.

**Institutional Review Board Statement:** Not applicable.

**Informed Consent Statement:** Not applicable.

**Data Availability Statement:** Not applicable.

**Acknowledgments:** We would like to thank Matthias Klingele from Fraunhofer Institute for Solar Energy Systems for the fruitful discussion.

**Conflicts of Interest:** The authors declare no conflict of interest.

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
