# Peer review of "Effect of Catalyst Ink and Formation Process on the Multiscale Structure of Catalyst Layers in PEM Fuel Cells"

_applsci, doi:10.3390/app12083776_

Round 1

Reviewer 1 Report

  1. Error in reference Figure and Table number.
    Line 494   Isn't it Figure 7, not Figure 5? 
    Line 665 Isn't it table 2, not Table 1?
  2. Japanese names are generally used as family names, so please change the following
    Line 1123 Takahiro should be changed to Suzuki
    Line 1146 Takahiro's should be changed to Suzuki's
    Line 1149 Takumi should be changed to Kusano

Author Response

We sincerely thank the reviewer for taking the time and effort to evaluate the manuscript. All the comments and suggestions raised have been addressed and the manuscript revised accordingly. Specifically,

Review comments:

Review comment #1: Error in reference Figure and Table number.

Line 494   Isn't it Figure 7, not Figure 5?

Line 665 Isn't it table 2, not Table 1?

 Authors’ response: The error for the figure number has been corrected in the revised manuscript. Please see Line 507 in the revised manuscript.

Line 665 (Line 679 of the revised manuscript): No, it is Table 1, not Table 2.

 Review comment #2: Japanese names are generally used as family names, so please change the following

Line 1123 Takahiro should be changed to Suzuki

Line 1146 Takahiro's should be changed to Suzuki's

Line 1149 Takumi should be changed to Kusano

Authors’ response: These typos and errors have been corrected in the revised manuscript. Please see Line 1139, Line 1162, and Line 1165 in the revised manuscript.

Reviewer 2 Report

Minor Revision

applsci-1669092-peer-review-v1

The submitted article “applsci-1669092-peer-review-v1” comprises the structure of catalyst layers (CLs) that significantly impacts the performance, durability, and cost of proton exchange membrane (PEM) fuel cells and is influenced by the catalyst ink and the CL formation process. The author mentioned to understand this relationship, promote the continuous and scalable production of membrane electrode assemblies, and guarantee the consistency of the CLs produced, further efforts need to be devoted to investigating the microstructure of catalyst ink (especially the catalyst ink with high solid content), the reversibility of the aged ink, and the drying process. Furthermore, except for the certain variables studied, the other manufacturing processes and conditions also require attention to avoid inconsistent conclusions.

However, apart from that, some points need to include and revise few sections of this review article, and thus a minor revision is required.

Comments are as follows:

Reviewer comments

  1. In the introduction section, the authors need to mention and cover-up microbial fuel cells for comparison with other reported approaches.

            Hence, fabrication, utilization, and application properties should be highlighted in the manuscript. These novel references must be included and highlighted:

doi:10.3390/en12030549
doi.org/10.3390/ma15010078
doi.org/10.1038/s41598-018-19617-2
DOI: 10.1038/srep17373
doi.org/10.1021/acs.jpcc.9b05105
doi.org/10.1021/acsanm.8b005488

  1. The introduction should be clarified in terms of uniqueness and the advantage of the novelty of this work over the previous related works.
  2. Figure 5. given information is not visible inside the figure, needs to revise and make readable. Explain the behavior of decreasing ratio effect of viscosity in the text with appropriate citation.
  3. Provide the full names for abbreviations when they appear for the first time in the text including "Abstract"
  4. Keywords are the words that best express the main content of the article, requiring refinement and accuracy. Please refine and confirm the keywords.
  5. The author needs to write a critical discussion with state-of-the-art
    literature after the presentation and discussion of your results.
  6. More elaboration and discussions should be given in the text about the finding and future perspectives of this work.
  7. The author should completely check and revise the format of their manuscript according to the author guidelines of this journal.

Overall, the writing and included information seem reasonable except for the above-mentioned comments, the author needs to include all the suggestions and cite the given articles before acceptance.

I would recommend that this review article should be accepted for publication in this journal after minor revision.

Author Response

We sincerely thank the reviewer for taking the time and effort to evaluate the manuscript. All the comments and suggestions raised have been addressed and the manuscript revised accordingly. Specifically,

Review comments:

The submitted article “applsci-1669092-peer-review-v1” comprises the structure of catalyst layers (CLs) that significantly impacts the performance, durability, and cost of proton exchange membrane (PEM) fuel cells and is influenced by the catalyst ink and the CL formation process. The author mentioned to understand this relationship, promote the continuous and scalable production of membrane electrode assemblies, and guarantee the consistency of the CLs produced, further efforts need to be devoted to investigating the microstructure of catalyst ink (especially the catalyst ink with high solid content), the reversibility of the aged ink, and the drying process. Furthermore, except for the certain variables studied, the other manufacturing processes and conditions also require attention to avoid inconsistent conclusions.

However, apart from that, some points need to include and revise few sections of this review article, and thus a minor revision is required.

Review comments:

Review comment #1: In the introduction section, the authors need to mention and cover-up microbial fuel cells for comparison with other reported approaches.

            Hence, fabrication, utilization, and application properties should be highlighted in the manuscript. These novel references must be included and highlighted:

doi:10.3390/en12030549

doi.org/10.3390/ma15010078

doi.org/10.1038/s41598-018-19617-2

DOI: 10.1038/srep17373

doi.org/10.1021/acs.jpcc.9b05105

doi.org/10.1021/acsanm.8b005488

 Authors’ response: The anodic and cathodic catalyst layers of proton exchange membrane fuel cell constitute the continuous and intersecting networks of Pt/C catalysts, proton-conductive ionomer, and pore. However, the anodic catalyst layer of microbial fuel cell constitutes biofilm of microorganism; the cathodic catalyst layer constitutes photocatalysts (e.g., TiO2) according to the references shown in the comments (doi.org/10.1038/s41598-018-19617-2; DOI: 10.1038/srep17373; doi.org/10.1021/acs.jpcc.9b05105). There is significant difference on the composition, structure, and fabrication process of catalyst layer between the proton exchange membrane fuel cell and microbial fuel cell. Therefore, microbial fuel cell is beyond the scope of the present review article, hence not mentioned.

For the articles suggested: The first article suggested has been cited in the manuscript. Please see reference 58. The second article suggested had been cited as reference 116 in the revised manuscript. The third, fourth, and fifth articles suggested are related to the microbial fuel cell so that they have not been cited in the revised manuscript. We did not find the last article suggested (DOI no found). Hence, it has not been added as a reference in the revised manuscript.

 Review comment #2: The introduction should be clarified in terms of uniqueness and the advantage of the novelty of this work over the previous related works.

Authors’ response: The uniqueness and advantage of this review manuscript have been clarified in Line 141-151 of the introduction section. To the best of authors’ knowledge, most reported reviews focus on the optimization of the formulation and dispersion techniques of catalyst ink, the characterization and modeling of catalyst ink, the coating techniques, or the underlying interactions between solvent, ionomer, and catalyst. Few reviews focus on the relationship between the composition and preparation process of the catalyst ink or the CL formation process and the multiscale structure of CL. Herein, our review focuses on the effect of the formulation and preparation of catalyst ink on its microstructure and macroscopic properties and further on the CL structure as well as the effect of the CL formation process on the multiscale structure of CL in order to clear the relationship of the recipe and preparation process of catalyst ink or the CL formation process and the multiscale structure of CL.

Review comment #3: Figure 5. given information is not visible inside the figure, needs to revise and make readable. Explain the behavior of decreasing ratio effect of viscosity in the text with appropriate citation.

Authors’ response: Figure 5 has been changed. Please see Line 321 in the revised manuscript.

For the behavior of decreasing ratio effect of viscosity: The related explanation has been added in the revised manuscript. Please see Line 316-318. Specifically,

The degree of shear thinning increases with increasing the concentration of catalyst or ionomer due to the increased degree of agglomeration [53], as shown in Figure 5b and c.

Review comment #4: Provide the full names for abbreviations when they appear for the first time in the text including "Abstract"

Authors’ response: We have rechecked all the abbreviation and made sure they have the full names for the first time.

Review comment #5: Keywords are the words that best express the main content of the article, requiring refinement and accuracy. Please refine and confirm the keywords.

Authors’ response: Thank you for the suggestion, and the keywords have been refined. Please see Line 24-25 in the revised manuscript.

Review comment #6: The author needs to write a critical discussion with state-of-the-art literature after the presentation and discussion of your results.

Authors’ response: Several discussions have been written in the manuscript. Please see Line 577-584, Line 786-797, and Line 1099-1104 in the manuscript. Some discussions have been added in the revised manuscript. Please see Line 450-458, Line 469-470, Line 485-487, and Line 827-829 in the revised manuscript.

Review comment #7: More elaboration and discussions should be given in the text about the finding and future perspectives of this work.

Authors’ response: More discussions have been added in the future direction and prospects. Please see Line 1238-1242 and Line 1293-1299 in the revised manuscript.

Review comment #8: The author should completely check and revise the format of their manuscript according to the author guidelines of this journal.

Authors’ response: We have carefully rechecked and revised the format of our manuscript based on the author guidelines of Applied Sciences.

Reviewer 3 Report

The presented review provides an extensive and interesting overview of the state of the art in the field of the catalyst Ink and formation process on the structure of Catalyst Layers in PEM Fuel Cells. In my opinion, the work thoroughly analyzes the problem and is well and clearly written. For this reason, I recommend it for publication.

Minor points:

‘The viscosity of a liquid always increases with temperature.’ –decreases

‘( ) with shear rate ( .’- strange symbols

‘The Newtonian fluid exhibits a linear function between the applied and the rate 285 of strain (or ).’ – the same as above

l.288 –L 292 the same problem

L487 ‘Besides, the I/C ratio has a vital effect on the microstructure and macroscopic properties of the catalyst ink, consequently the CL structure.’-what do you mean?

Author Response

We sincerely thank the reviewer for taking the time and effort to evaluate the manuscript. All the comments and suggestions raised have been addressed and the manuscript revised accordingly. Specifically,

Review comments:

The presented review provides an extensive and interesting overview of the state of the art in the field of the catalyst Ink and formation process on the structure of Catalyst Layers in PEM Fuel Cells. In my opinion, the work thoroughly analyzes the problem and is well and clearly written. For this reason, I recommend it for publication.

Review comments:

Review comment #1: ‘The viscosity of a liquid always increases with temperature.’ –decreases

 Authors’ response: The error has been corrected in the revised manuscript. Please see Line 271 in the revised manuscript.

Review comment #2: ‘( ) with shear rate ( .’- strange symbols

‘The Newtonian fluid exhibits a linear function between the applied and the rate 285 of strain (or ).’ – the same as above

l.288 –L 292 the same problem

Authors’ response: These typos and errors have been corrected in the revised manuscript. Please see Line 282-292 in the revised manuscript.

 Review comment #3: L487 ‘Besides, the I/C ratio has a vital effect on the microstructure and macroscopic properties of the catalyst ink, consequently the CL structure.’-what do you mean?

Authors’ response: This sentence has been rewritten in the revised manuscript (Line 500-502). Specifically,

‘Besides, the I/C ratio has a vital effect on the microstructure and macroscopic properties of the catalyst ink and further significantly impacts the CL structure.’

Reviewer 4 Report

This article reviews the latest reports mainly within the past five years on the constituent materials, composition, preparation method of the catalyst ink for the PEMFC, and application process and drying process of the catalyst ink to MEA. The authors summarize this review from the view point of the multiscale structure of the catalyst layer which is a key factor to prepare a highly active and efficient PEMFCs in practical. The composition of the manuscript divided into catalyst ink and CL formation is appropriate and well organized. It would be a useful review for the reader. The following minor modifications are required.

The titles of sections 3 and 4 are the same and should be corrected.

The heading “5)” at line 369 would be “3)”.

Author Response

We sincerely thank the reviewer for taking the time and effort to evaluate the manuscript. All the comments and suggestions raised have been addressed and the manuscript revised accordingly. Specifically,

Review comments:

This article reviews the latest reports mainly within the past five years on the constituent materials, composition, preparation method of the catalyst ink for the PEMFC, and application process and drying process of the catalyst ink to MEA. The authors summarize this review from the view point of the multiscale structure of the catalyst layer which is a key factor to prepare a highly active and efficient PEMFCs in practical. The composition of the manuscript divided into catalyst ink and CL formation is appropriate and well organized. It would be a useful review for the reader. The following minor modifications are required.

Review comments:

Review comment #1: The titles of sections 3 and 4 are the same and should be corrected.

 Authors’ response: The error has been corrected in the revised manuscript. Please see Line 397 in the revised manuscript.

Review comment #2: The heading “5)” at line 369 would be “3)”.

Authors’ response: The error has been corrected in the revised manuscript. Please see Line 371 in the revised manuscript.
